# Robust T cell activation requires an eIF3-driven burst in T cell receptor translation

Dasmanthie De Silva[1,2], Lucas Ferguson[1], Grant H Chin[1], Benjamin E Smith[3], Ryan A Apathy[4], Theodore L Roth[4], Franziska Blaeschke[5], Marek Kudla[1], Alexander Marson[4,5,6,7,8,9,10], Nicholas T Ingolia[1,11], Jamie HD Cate[1,2,10,11,12,13]*

[1]Department of Molecular and Cell Biology, University of California-Berkeley, Berkeley, United States; [2]The J. David Gladstone Institutes, San Francisco, United States; [3]School of Optometry, University of California, Berkeley, Berkeley, United States; [4]Department of Microbiology and Immunology, University of California, San Francisco, San Francisco, United States; [5]Gladstone-UCSF Institute of Genomic Immunology, San Francisco, United States; [6]Diabetes Center, University of California, San Francisco, San Francisco, United States; [7]Chan Zuckerberg Biohub, San Francisco, United States; [8]Department of Medicine, University of California, San Francisco, San Francisco, United States; [9]Parker Institute for Cancer Immunotherapy, San Francisco, United States; [10]Innovative Genomics Institute, University of California, Berkeley, Berkeley, United States; [11]California Institute for Quantitative Biosciences, University of California, Berkeley, Berkeley, United States; [12]Department of Chemistry, University of California-Berkeley, Berkeley, United States; [13]Molecular Biophysics and Integrated Bioimaging Division, Lawrence Berkeley National Laboratory, Berkeley, United States

*For correspondence:
j-h-doudna-cate@berkeley.edu

**Abstract** Activation of T cells requires a rapid surge in cellular protein synthesis. However, the role of translation initiation in the early induction of specific genes remains unclear. Here, we show human translation initiation factor eIF3 interacts with select immune system related mRNAs including those encoding the T cell receptor (TCR) subunits TCRA and TCRB. Binding of eIF3 to the *TCRA* and *TCRB* mRNA 3'-untranslated regions (3'-UTRs) depends on CD28 coreceptor signaling and regulates a burst in TCR translation required for robust T cell activation. Use of the *TCRA* or *TCRB* 3'-UTRs to control expression of an anti-CD19 chimeric antigen receptor (CAR) improves the ability of CAR-T cells to kill tumor cells in vitro. These results identify a new mechanism of eIF3-mediated translation control that can aid T cell engineering for immunotherapy applications.

## Editor's evaluation

The work addresses the role of translation initiation in the early induction of specific genes during T cell activation. It primarily uses the Jurkat T cell line and demonstrates that the translation initiation factor eIF3 interacts with the 3'-untranslated regions of the TCRA and TCRB mRNAs. This interaction resulted in a rapid burst in TCRA and TCRB translation through a mechanism dependent on CD28. Adding the TCRA or TCRB 3' UTR to an anti-CD19 chimeric antigen receptor (CAR) improves the ability of the corresponding CAR-T cells to kill tumor cells in vitro.

## Introduction

Translation initiation serves as a key gatekeeper of protein synthesis in eukaryotes and requires the action of eukaryotic initiation factor 3 (eIF3) (*Hernández et al., 2020*; *Pelletier and Sonenberg, 2019*). In humans, eIF3 is a 13-subunit protein complex that coordinates the cellular machinery in positioning ribosomes at the mRNA start codon. Several lines of evidence indicate eIF3 also serves specialized roles in cellular translation, by recognizing specific RNA structures in the 5'-untranslated regions (5'-UTRs) of target mRNAs (*Lee et al., 2015*), binding the 7-methyl-guanosine (m$^7$G) cap (*Lamper et al., 2020*; *Lee et al., 2016*) or through interactions with *N*-6-methyl-adenosine (m$^6$A) post-transcriptional modifications in mRNAs (*Meyer et al., 2015*). Binding to these *cis*-regulatory elements in mRNA can lead to translation activation or repression, depending on the RNA sequence and structural context (*de la Parra et al., 2018*; *Lee et al., 2016*; *Lee et al., 2015*; *Meyer et al., 2015*). These non-canonical functions for eIF3 can aid cell proliferation (*Lee et al., 2015*), or allow cells to rapidly adapt to stresses such as heat shock (*Meyer et al., 2015*). In the immune system, T cell activation requires a rapid increase in protein synthesis within the first few hours that also involves eIF3 (*Ahern et al., 1974*; *Kleijn and Proud, 2002*; *Miyamoto et al., 2005*; *Ricciardi et al., 2018*). However, whether eIF3 serves a general or more specific role in T cell activation is unknown.

Translation in non-activated lymphocytes is limited by the availability of translation initiation factors (*Ahern et al., 1974*; *Mao et al., 1992*; *Wolf et al., 2020*). The mRNAs for several translation initiation factors, including those for many eIF3 subunits, are repressed in resting T cells and are rapidly translated within hours of activation (*Wolf et al., 2020*). Additionally, nearly inactive eIF3 in quiescent T cells is activated to form translation initiation complexes in the first few hours after stimulation (*Miyamoto et al., 2005*). Activation of eIF3 coincides with the recruitment of subunit eIF3j to eIF3 and translation initiation complexes (*Miyamoto et al., 2005*). Post-translational modifications are also thought to contribute to the early increase in translation, for example activation of the guanine nucleotide exchange factor eIF2B (*Kleijn and Proud, 2002*). By contrast, the level of the canonical mRNA cap-binding complex eIF4F–composed of translation initiation factors eIF4E, eIF4G, and eIF4A–increases much later after T cell activation (*Mao et al., 1992*).

Recent studies have identified translational control of specific transcripts required for the rewiring of metabolism needed for effector T cell function. Translation of these specific transcripts involved in glycolysis and fatty acid synthesis occurs within 1–3 days of T cell activation (*Ricciardi et al., 2018*), and depends on the activity of eIF4F. Hundreds of other mRNAs translationally repressed in resting T cells are rapidly translated within hours after activation, including those encoding key transcription factors and ribosomal proteins, in addition to the translation initiation factors noted above (*Wolf et al., 2020*). Most of these mRNAs are sensitive to mTOR inhibition, highlighting the importance of eIF4F for the translation of these specific transcripts. However, a small subset of mRNAs evade mTOR inhibition by an unknown mechanism (*Wolf et al., 2020*). Whether eIF3 selectively regulates these or other mRNAs early in T cell activation is not known.

In the adaptive immune system, T cells are activated when a foreign antigen is recognized by the T cell receptor (TCR). However, robust T cell activation requires a second signal generated by interactions between the T cell and antigen presenting cell mediated by the co-receptor CD28 (*Esensten et al., 2016*). This two-signal mechanism enables T cells to adopt a fully active state and avoid becoming unresponsive (*Chen and Flies, 2013*). The process of T cell activation can be mimicked in vitro using antibodies targeting both the TCR and CD28 (anti-CD3 and anti-CD28 antibodies, respectively) (*Harding et al., 1992*), which has greatly aided the dissection of molecular mechanisms underlying T cell activation.

Insights from these studies have also inspired efforts to engineer T cells for immunotherapy applications such as treating cancers (*Chen and Flies, 2013*). T cells can be engineered to express chimeric antigen receptors (CARs) that specifically target antigens on the surface of cancer cells, and signal through protein elements derived from both the TCR and CD28 or other co-stimulatory receptors (*Chen and Flies, 2013*; *Globerson Levin et al., 2021*). The most successful of these CAR T cells have been used to treat CD19-positive B cell malignancies, with dramatic results (*Friedman et al., 2018*; *Kalos et al., 2011*; *Kochenderfer et al., 2013*; *Qin et al., 2020*; *Wang et al., 2020*). However, CAR T cells still have a number of drawbacks, including toxicity to the patient, CAR T cell exhaustion, limited persistence and poor efficacy in solid tumors. These problems highlight the need for a deeper

understanding of T cell activation and how it can be controlled in CAR T cells (*Globerson Levin et al., 2021*; *Watanabe et al., 2018*).

Here, we identified mRNAs that specifically bind eIF3 in activated T cells, including many encoding proteins involved in immune cell function such as the TCR. We mapped the eIF3-dependent *cis*-regulatory elements in the mRNAs encoding the TCR alpha and beta subunits (TCRA and TCRB, respectively), finding the 3'-untranslated regions (3'-UTRs) of these mRNAs control a rapid burst in TCRA and TCRB translation that depends on CD28 coreceptor signaling. Finally, we use this information to engineer T cells expressing chimeric antigen receptors to modulate the dynamics of CAR expression and improve the ability of CAR T cells to kill tumor cells in vitro.

## Results

### A specific suite of RNAs interact with eIF3 in activated Jurkat cells

To delineate how eIF3 contributes to T cell activation, we first identified RNAs that directly interact with eIF3 in Jurkat cells activated for 5 hours with phorbol 12-myristate 13-acetate and ionomycin (PMA + I), using photoactivatable ribonucleoside-enhanced crosslinking and immunoprecipitation (PAR-CLIP) (*Hafner et al., 2010*; *Lee et al., 2015*; *Mukherjee et al., 2019*; *Figure 1A*). In the Jurkat PAR-CLIP experiments, RNA crosslinked to eight of the 13 eIF3 subunits, as identified by mass spectrometry: subunits EIF3A, EIF3B, EIF3D, and EIF3G as previously seen in HEK293T cells (*Lee et al., 2015*), as well as subunits EIF3C, EIF3E, EIF3F, and EIF3L (*Figure 1B*, *Figure 1—figure supplement 1A–1B*, *Supplementary file 1*). Consistent with its role in T cell activation, eIF3 crosslinked to a substantially larger number of mRNAs (~75 x more) in activated Jurkat cells compared to control non-activated cells (*Figure 1—figure supplement 1C–1D* and *Supplementary files 2 and 3*). Notably, in activated Jurkat cells eIF3 interacted with mRNAs enriched for those encoding proteins central to immune cell function, in contrast to those previously identified in HEK293T cells (*Lee et al., 2015*; *Figure 1C* and *Figure 1—figure supplement 1E*). The extent of eIF3 crosslinking in activated Jurkat cells does not correlate with mRNA abundance based on transcription profiling carried out in parallel (*Figure 1—figure supplement 1F* and *Supplementary file 4*). This suggests the enrichment of immune system related mRNAs reflects the involvement of eIF3 in specific regulation of T cell activation.

In activated Jurkat cells, eIF3 showed a multitude of crosslinking patterns on different mRNAs (*Figure 1D–E* and *Figure 1—figure supplement 1D, G and H*), consistent with varied roles for eIF3 in T cell activation and function. Many of the mRNAs have a single PAR-CLIP site in the 5'-UTR as observed in HEK293T cells (*Lee et al., 2015*; *Figure 1D*, *Figure 1—figure supplement 1D*, *Supplementary file 3*). However, eIF3 crosslinked to some mRNAs across the entire length of the transcript, from the beginning of the 5'-UTR through the 3'-UTR (*Figure 1E* and *Figure 1—figure supplement 1H*). This 'pan-mRNA' pattern of eIF3 crosslinking–which includes polyadenylated mRNAs as well as histone mRNAs–has not been observed before. Interestingly, a number of these mRNAs encode proteins important for T cell activation, including both the alpha and beta subunits of the T cell receptor (TCR), subunits TCRA and TCRB (*Figure 1E*, *Supplementary file 5*).

### *TCRA* and *TCRB* mRNAs do not colocalize with P bodies or stress granules in activated Jurkat cells

Crosslinking in PAR-CLIP experiments requires direct interaction between the RNA and protein of interest (*Ascano et al., 2012*). Thus, the pan-mRNA pattern of crosslinking between eIF3 and certain mRNAs suggests formation of ribonucleoprotein complexes (RNPs) highly enriched in eIF3. Notably, the pan-mRNA crosslinking pattern in the *TCRA* and *TCRB* mRNAs occurs in activated but not in non-activated Jurkat cells (*Figure 1E*), suggesting eIF3 may contribute to increased translation of these mRNAs rather than their repression. We therefore examined *TCRA* and *TCRB* mRNA localization in activated Jurkat cells, to determine whether they colocalized with repressive environments such as P bodies or stress granules (*Tauber et al., 2020*). Since Jurkat cells have a defined TCR, we designed fluorescence in situ hybridization (FISH) probes across the entire *TCRA* and *TCRB* transcripts to examine their localization. Interestingly, the *TCRA* and *TCRB* mRNAs did not co-localize with either P bodies or stress granules, or with each other in Jurkat cells activated with anti-CD3/anti-CD28 antibodies, which induce both TCR and CD28 coreceptor signaling required for robust T cell activation (*Harding et al., 1992*; *Figure 1F*, *Figure 1—figure supplement 2A and B*). These results suggest

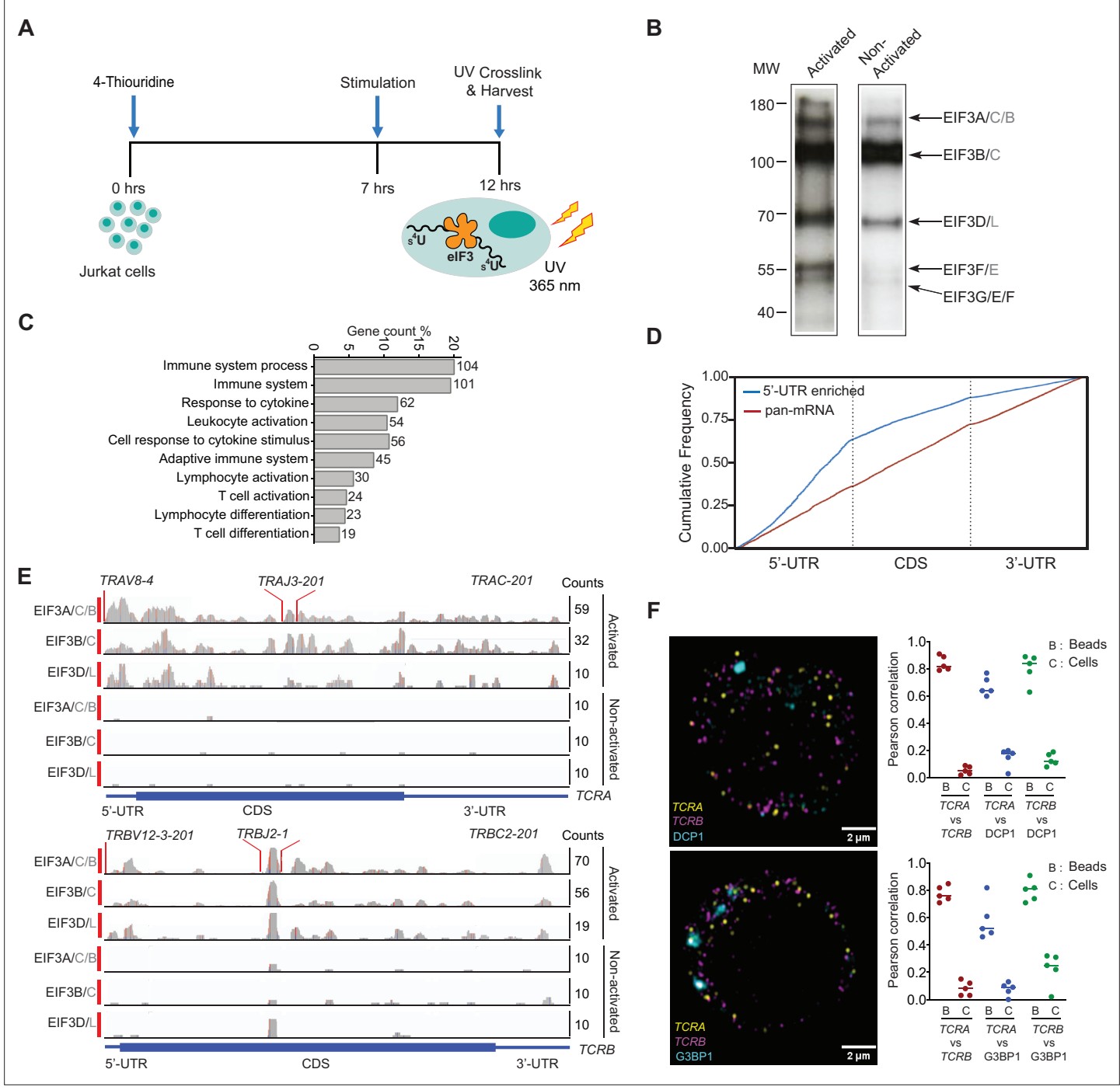

**Figure 1.** eIF3 interacts with specific mRNAs related to immune function in activated Jurkat cells. (**A**) Schematic of the PAR-CLIP experiment in Jurkat cells, showing steps through cell harvesting. (**B**) Phosphorimage of SDS polyacrylamide gel resolving 5' $^{32}$P-labeled RNAs crosslinked to eIF3 subunits in activated and non-activated Jurkat cells, from one of two biological replicates. (**C**) Pathway enrichment categories determined using the STRING Database for both biological replicates of the EIF3A/C/B PAR-CLIP libraries (mRNAs with ≥ 100 reads). Number of genes in each pathway whose mRNAs crosslinked to eIF3 is shown next to each bar. Note that the categories reported by the STRING Database are not disjoint sets. (**D**) Varied mRNA crosslinking patterns to eIF3 in activated Jurkat cells. Cumulative plot showing mRNA crosslinking in sample EIF3A/C/B to predominantly the 5'-UTR (n = 396, 414 mRNAs in replicates 1 and 2, respectively), and across the entire length of some mRNAs ('pan-mRNAs', n = 634, 621 mRNAs). The 5'-UTR, CDS, and 3'-UTR regions are shown normalized by length. (**E**) Crosslinking of the eIF3 subunits as indicated across the entire *TCRA* and *TCRB* mRNAs, in activated and non-activated Jurkat cells. 5'-UTR, coding sequence (CDS), and 3'-UTR elements (below) along with the variable (**V**), joining (**J**) and constant (**C**) regions (above) for the mapped TCR genes in Jurkat cells are shown. The blue and red vertical lines in the plotted reads indicate the amount of T-C transitions vs other mutations, respectively for a particular nucleotide. The *TCRA* and *TCRB* mRNAs are present in both non-activated

*Figure 1 continued on next page*

*Figure 1 continued*

and activated Jurkat cells (**Supplementary file 4**). (**F**) FISH analysis of *TCRA* and *TCRB* mRNAs (yellow and magenta, respectively) and P bodies (top) marked by the location of DCP1 (cyan) and stress granules (bottom) marked by the location of G3BP1 (cyan), in activated Jurkat cells. Graphs to the right of the images indicate Pearson's correlation coefficients (PCCs) of *TCRA* and *TCRB* mRNAs localizing with each other or with P bodies or stress granules. TetraSpeck microsphere beads were used as a positive control for colocalization. Labels on the x axis are, B: TetraSpeck microsphere beads, C: activated Jurkat cells. (n = 5, p < 0.008, for PCC values of cells relative to bead colocalization, across all the channels tested, using the Wilcoxon rank-sum test). Images are representative of one experiment of the five independent experiments in the graphs.

The online version of this article includes the following figure supplement(s) for figure 1:

**Figure supplement 1.** eIF3 PAR-CLIP experiments in activated and non-activated Jurkat cells.

**Figure supplement 2.** The *TCRA* and *TCRB* mRNAs do not colocalize with P bodies and stress granules in activated T cells.

---

that the *TCRA* and *TCRB* mRNAs are not translationally repressed but are possibly activated by eIF3 binding.

## Pan-mRNAs remain bound to eIF3 in translating ribosomes

The pan-mRNA crosslinking pattern suggests that eIF3 remains bound to the *TCRA* and *TCRB* mRNAs while they are actively translated. To capture *TCRA* and *TCRB* mRNAs in translating ribosomes and examine their interactions with eIF3, we analyzed polysomes in Jurkat cells activated with anti-CD3/anti-CD28 antibodies (**Figure 2A**; *Harding et al., 1992*). The cells were first treated with protein-protein crosslinker dithiobis(succinimidyl propionate) (DSP) before isolating polysomes on sucrose gradients (**Figure 2—figure supplement 1A**). We then incubated the cell lysates with RNase H and DNA oligonucleotides designed to cleave the mRNAs specifically between the 5'-UTR, coding sequence (CDS), and 3'-UTR (**Figure 2B**, **Supplementary file 6**). This protocol efficiently cleaved the mRNAs and prevented the released 5'-UTR and 3'-UTR elements from entering polysomes (**Figure 2—figure supplement 1B–1E**). It also allowed us to detect eIF3 interactions with the mRNA CDS regions independent of eIF3 interactions with the UTR sequences identified in the PAR-CLIP analysis. We detected mRNAs interacting with eIF3 in the polysomes by performing anti-EIF3B immunoprecipitations followed by qRT-PCR (**Figure 2A**). We compared both *TCRA* and *TCRB* mRNAs to another pan-crosslinked mRNA, *DUSP2*, and to two mRNAs that crosslinked to eIF3 only through their 5'-UTRs (*EGR1*, *TRIM28*). Using primers to the CDS regions of the mRNAs, we found that eIF3 only immunoprecipitated the pan-crosslinked mRNAs (*TCRA*, *TCRB*, *DUSP2*) from polysomes, but not mRNAs that only crosslinked to eIF3 through their 5'-UTRs (*EGR1*, *TRIM28*) (**Figure 2C and D**). Importantly, all these mRNAs are present in translating ribosomes and can be immunoprecipitated with eIF3 when the mRNAs are left intact (RNase H treatment without DNA oligos) (**Figure 2E**).

We also tested whether these mRNAs interact with eIF3 similarly in primary human T cells during activation. We could not examine *TCRA* and *TCRB* mRNAs in primary human T cells, as these mRNAs do not have a unique sequence in the 5'-UTR or variable region of the CDS to which we could design DNA oligonucleotides as described above. Therefore, we tested the distribution of *DUSP2*, *EGR1*, and *TRIM28* mRNAs in primary human T cells activated with anti-CD3/anti-CD28 antibodies (*Harding et al., 1992*). As observed in Jurkat cells, *DUSP2* mRNA remained bound to eIF3 through its CDS region in polysomes whereas the *EGR1* and *TRIM28* mRNAs did not remain bound (**Figure 2F–H**). We also confirmed that the 5'-UTR and 3'-UTR elements of these mRNAs are efficiently cleaved and did not enter polysomes (**Figure 2—figure supplement 1F–1H**). Taken together, these results indicate that, in activated T cells, eIF3 remains bound to the coding sequences (CDS) of the pan-mRNAs *TCRA*, *TCRB*, and *DUSP2* in polysomes independent of their 5'-UTR and 3'-UTR elements. These results further support the model that the pan-mRNA crosslinking pattern of *TCRA* and *TCRB* mRNAs reflects eIF3 binding to these mRNAs during translation elongation.

## eIF3 interacts with the *TCRA* and *TCRB* mRNA 3'-UTRs and controls a burst in translation during T cell activation

Although crosslinking of eIF3 to the CDS regions of the *TCRA* and *TCRB* mRNAs (**Figure 1E**) indicates eIF3 remains bound to them during translation elongation (**Figure 2** and **Figure 2—figure supplement 1**), we wondered whether the 5'-UTRs and 3'-UTRs might play a role in recruiting these mRNAs to translating ribosomes. The *TCRA* and *TCRB* genes encode a different, often short 5'-UTR for each

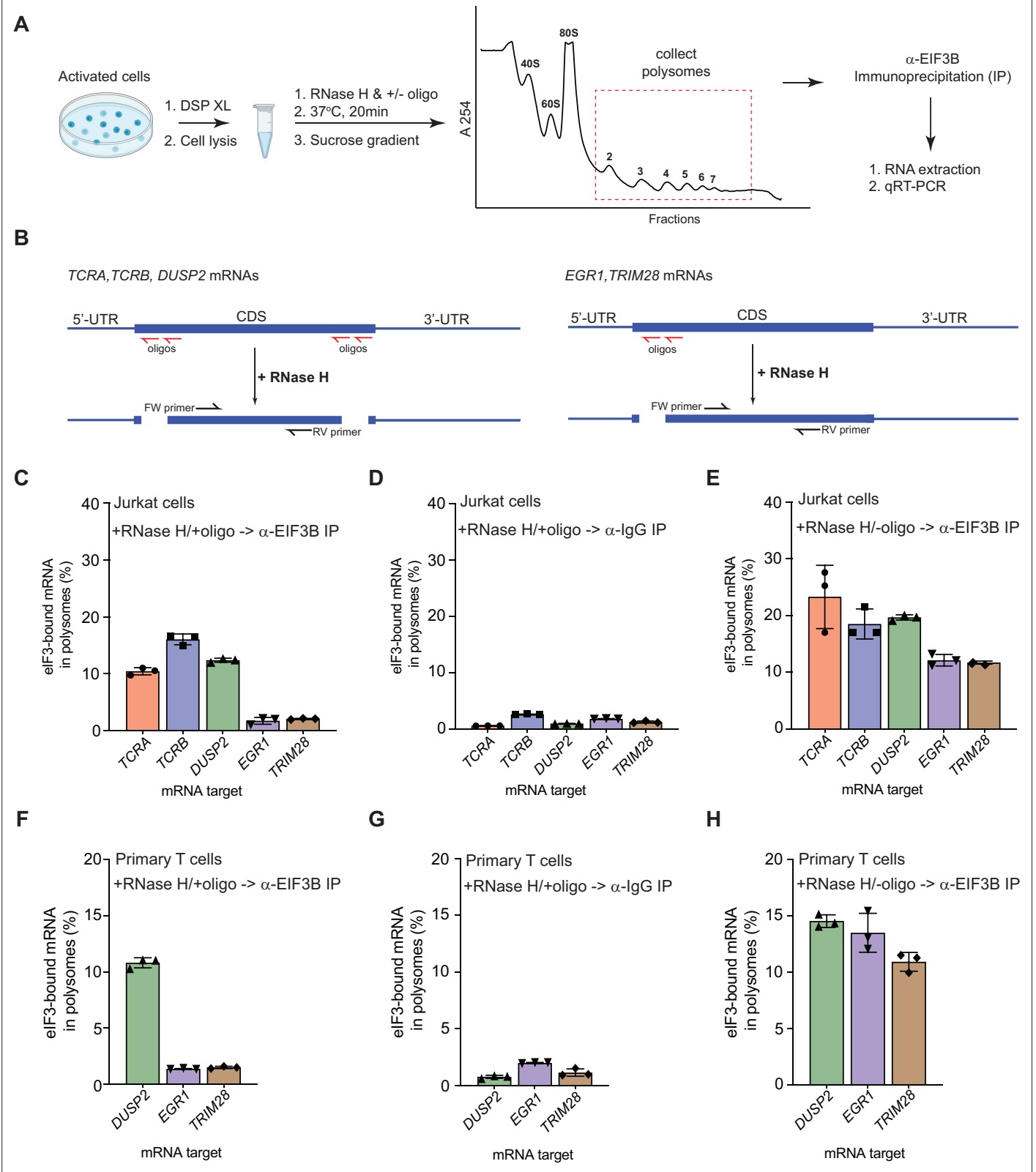

**Figure 2.** eIF3 remains bound to the coding sequences (CDS) of pan-mRNAs independent of their 5'-UTR and 3'-UTR elements in actively translating ribosomes. (**A**) Schematic outlining the RNase H-based assay of eIF3 interactions with mRNAs in polysomes. DSP refers to the dithiobis (succinimidyl propionate) crosslinking agent. Oligos, DNA oligos designed for RNase H-mediated targeting and cleavage of specific mRNAs. (**B**) Strategy for detecting mRNA fragments released by RNase H digestion. Red arrows denote DNA oligos for RNase H-mediated targeting of mRNAs. RT-qPCR

*Figure 2 continued on next page*

*Figure 2 continued*

primers (black) were used to detect the CDS regions of the mRNAs. (**C**) Amount of eIF3-bound mRNA co-immunoprecipitated by an anti-EIF3B antibody (*Lee et al., 2015*), from polysome fractions of Jurkat cells treated with RNase H and oligos targeting the CDS-UTR junctions (red arrows diagrammed in panel **B**). (**D**) Amount of eIF3-bound mRNA co-immunoprecipitated with IgG beads, from polysome fractions of Jurkat cell lysate treated with RNase H and oligos targeting the CDS-UTR junctions. (**E**) Amount of eIF3-bound mRNA co-immunoprecipitated by the anti-EIF3B antibody, from polysome fractions of Jurkat cell lysate treated only with RNase H. (**F**) Amount of eIF3-bound mRNA co-immunoprecipitated by an anti-EIF3B antibody, from polysome fractions of primary human T cells treated with RNase H and oligos targeting the CDS-UTR junctions (red arrows diagrammed in panel **B**). (**G**) Amount of eIF3-bound mRNA co-immunoprecipitated with IgG beads, from polysome fractions of primary human T cell lysate treated with RNase H and oligos targeting the CDS-UTR junctions. (**H**) Amount of eIF3-bound mRNA co-immunoprecipitated by the anti-EIF3B antibody, from polysome fractions of primary human T cell lysate treated only with RNase H. In panels C–H, the percentage is relative to the amount of total mRNA present in the polysome fraction prior to immunoprecipitation. All the immunoprecipitation experiments in panels **C**–**H** were carried out in biological duplicate with one technical triplicate shown (n = 3, with mean and standard deviations shown). The primary human T cell experiment was done using two donors.

The online version of this article includes the following figure supplement(s) for figure 2:

**Figure supplement 1.** The *TCRA* and *TCRB* mRNAs remain bound to elongating ribosomes via eIF3 in activated T cells.

variable region of the mature locus (*Omer et al., 2021*; *Scaviner and Lefranc, 2000*), suggesting the 5'-UTR is unlikely to harbor eIF3-dependent regulatory elements. We therefore focused on the roles of the *TCRA* and *TCRB* 3'-UTRs. We constructed nanoluciferase reporters fused to the WT *TCRA* or *TCRB* mRNA 3'-UTR sequence, to 3'-UTRs with the eIF3 PAR-CLIP site deleted (*ΔPAR*) or to 3'-UTRs with the reversed sequence of the eIF3 PAR-CLIP site (R\**PAR*, i.e. 5'–3' sequence reversed to 3'–5' direction to maintain the length of the 3'-UTR) (*Figure 3A*). We then stably expressed these mRNAs in primary human T cells using lentiviral transduction and activated these T cells using anti-CD3/anti-CD28 antibodies. T cells expressing the reporters with the WT *TCRA* or *TCRB* mRNA 3'-UTR sequences produced substantially higher luminescence that peaked 1 hr after activation, while cells expressing nanoluciferase from reporters with a deletion or reversal of the eIF3 PAR-CLIP site sequence showed no apparent burst in translation (*Figure 3B*). The *TCRA ΔPAR* or R\**PAR* or *TCRB ΔPAR* or R\**PAR* mutations, however, did not cause significant defects in the nanoluciferase mRNA levels when compared to reporters with the corresponding WT 3'-UTR sequences (*Figure 3C*). This suggests the burst in nanoluciferase expression observed with the WT *TCRA* or *TCRB* mRNA 3'-UTR sequences is regulated posttranscriptionally. Immunoprecipitation of eIF3 followed by qRT-PCR quantification of nanoluciferase mRNA showed that less nanoluciferase mRNA bound to eIF3 when the 3'-UTR PAR-CLIP site was either deleted or reversed, compared to nanoluciferase mRNAs carrying the WT *TCRA* or *TCRB* 3'-UTR (*Figure 3D and E*). Interestingly, although the *TCRA* and *TCRB* mRNA 3'-UTR elements that crosslinked to eIF3 do not share any conserved sequences or RNA structural elements (*Xu and Mathews, 2016*), the above results support the idea that eIF3 binding to specific sequences or structures within the *TCRA* and *TCRB* 3'-UTRs controls a burst in translation after T cell activation.

We next tested whether the *TCRA* and *TCRB* 3'-UTRs serve to initially recruit eIF3 to the *TCRA* and *TCRB* mRNAs, as eIF3 does not remain stably bound to these 3'-UTRs in translating ribosomes (*Figure 1—figure supplement 2G*). We made nanoluciferase reporters in which we replaced the eIF3 PAR-CLIP site in the *TCRA* 3'-UTR with sequences from the hepatitis C viral internal ribosome entry site (HCV IRES domain IIIabc) or the *JUN* mRNA 5'-UTR, each previously shown to bind directly to eIF3 (*Kieft et al., 2001*; *Lee et al., 2015*; *Figure 3F*). We stably transduced these constructs into primary T cells and measured nanoluciferase activity after activating the cells with anti-CD3/anti-CD28 antibodies. By contrast to the *TCRA ΔPAR* 3'-UTR, the HCV IRES IIIabc and *JUN* sequences increased nanoluciferase translation (*Figure 3G*). This was true despite the fact that there was less of an increase in mRNA abundance for the HCV IRES IIIabc and *JUN* constructs compared to the *TCRA ΔPAR* 3'-UTR (*Figure 3H*). The HCV IRES IIIabc and *JUN* sequences also rescued eIF3 binding to the reporter mRNAs (*Figure 3I*) upon T cell activation. These results are consistent with the engineered 3'-UTRs recruiting eIF3 to the mRNA. However, the dynamics of translation activation induced by the 3'-UTRs harboring the HCV IRES or *JUN* sequences did not recapitulate the effects of the WT *TCRA* 3'-UTR. Both engineered 3'-UTRs increased nanoluciferase levels within 30 min of activation, in contrast to the WT *TCRA* 3'-UTR (*Figure 3G*). Furthermore, neither engineered 3'-UTR led to a marked decrease of nanoluciferase levels after the 1 hr peak in luminescence seen with the WT *TCRA* 3'-UTR (*Figure 3G*). Taken together, the nanoluciferase reporter experiments reveal the *TCRA* and *TCRB* mRNA 3'-UTRs

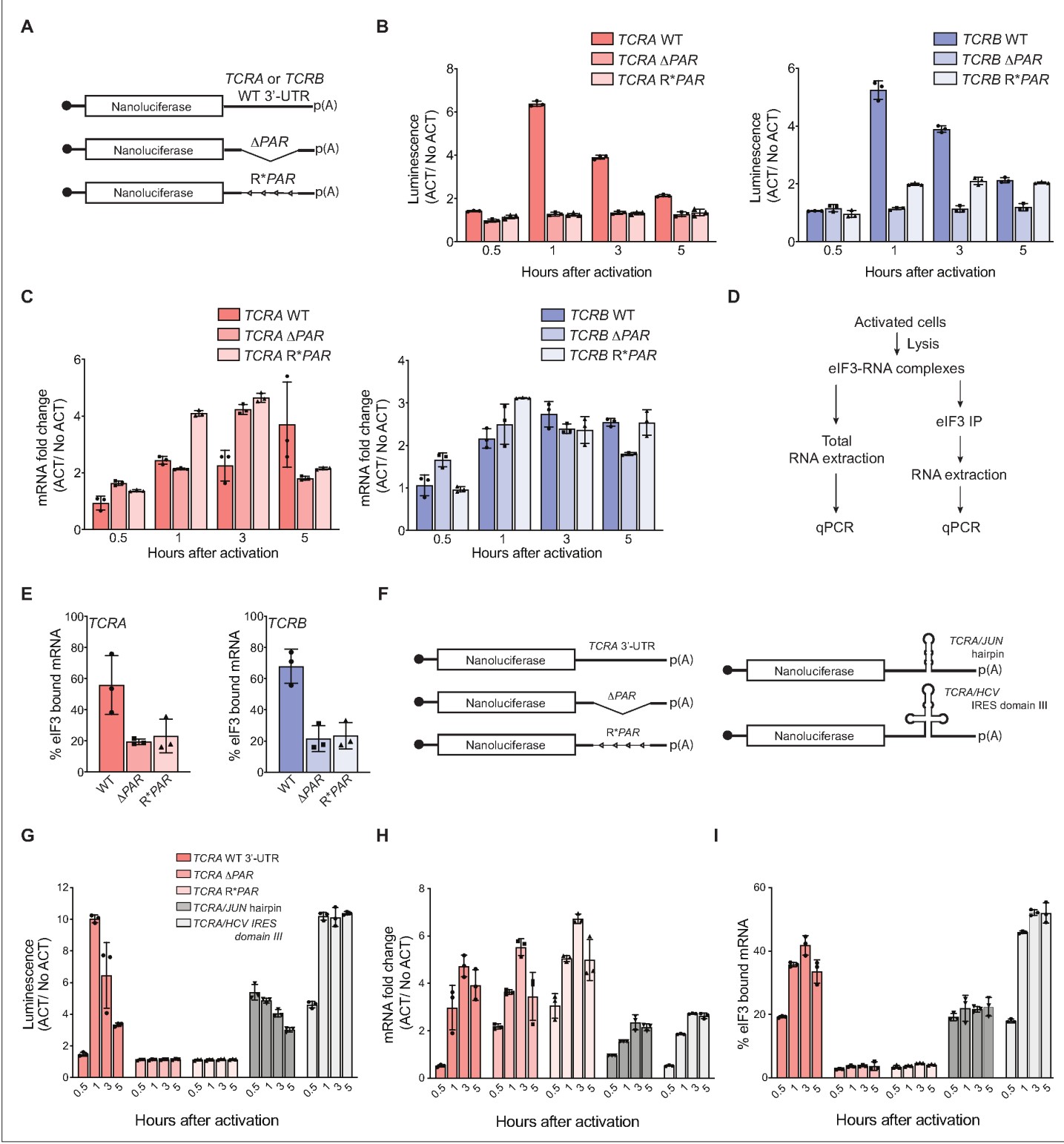

**Figure 3.** Interaction of eIF3 with *TCRA* and *TCRB* mRNA 3'-UTR elements mediates a burst in TCR translation in primary human T cells. (**A**) Schematic of nanoluciferase reporters stably expressed in primary human T cells. The reporters carry the *HBB* 5'-UTR and WT, *ΔPAR* or R**PAR* 3'-UTRs. WT, intact 3'-UTR from either *TCRA* or *TCRB* mRNA; *ΔPAR*, 3'-UTR of *TCRA* or *TCRB* with the eIF3 PAR-CLIP site deleted; R**PAR*, reversed PAR-CLIP sequence in the 3'-UTR of *TCRA* or *TCRB* mRNA. (**B**) Luciferase activity in anti-CD3/anti-CD28 activated T cells stably expressing nanoluciferase constructs described in **A**, relative to non-activated controls (ACT/NoACT). (**C**) Changes in nanoluciferase mRNA levels in **B**, as determined by qRT-PCR. (**D**) Schematic of immunoprecipitation of eIF3 using an anti-EIF3B antibody (***Lee et al., 2015***), followed by qRT-PCR to quantify the amount of nanoluciferase mRNA

*Figure 3 continued on next page*

*Figure 3 continued*

bound to eIF3. (**E**) Immunoprecipitation as shown in **D** showing the amount of nanoluciferase mRNA bound to eIF3 in T cells stably transduced with either *TCRA* (left) or *TCRB* (right) WT, *ΔPAR* or R*PAR* 3'-UTRs after activation with anti-CD3/anti-CD28 antibodies for 1 hr. The percent mRNA bound to anti-EIF3B beads is calculated relative to total mRNA isolated from the cells. (**F**) Schematic of nanoluciferase reporters stably expressed in primary human T cells. Nanoluciferase reporters carry the *HBB* 5'-UTR and WT, *ΔPAR*, R*PAR* of the *TCRA* 3'-UTR, or *ΔPAR* of the *TCRA* 3'-UTR replaced with either a *JUN* 5'-UTR hairpin or hepatitis C viral (HCV) internal ribosome entry site (domain IIIabc). (**G**) Luciferase activity in anti-CD3/anti-CD28 activated T cells stably expressing nanoluciferase constructs described in **F**, relative to non-activated controls. (**H**) Changes in nanoluciferase mRNA levels in **G**, as determined by qRT-PCR. (**I**) Immunoprecipitation as shown in **D** to quantify the amount of nanoluciferase mRNA bound to eIF3 in T cells stably expressing nanoluciferase constructs in **F** after activation with anti-CD3/anti-CD28 antibodies for 0.5, 1, 3, and 5 hr. The percent mRNA bound to anti-EIF3B beads is calculated relative to total mRNA isolated from the cells. All experiments were carried out in triplicate (three separate wells per condition), with mean and standard deviations shown. All the primary human T cell experiments were performed using two donors and data from one representative donor is shown.

are necessary and sufficient to drive a burst in translation after T cell activation. They also suggest these 3'-UTR elements recruit eIF3 to the *TCRA* and *TCRB* mRNAs to drive a burst in translation of the TCR alpha and beta subunits.

## The *TCRA* and *TCRB* mRNA 3'-UTRs control a burst in translation in a CD28-dependent manner

Robust T cell activation after antigen recognition generally requires signaling both through the TCR and through the costimulatory receptor CD28, which interacts with proteins CD80 and CD86 on the surface of antigen presenting cells (*Adams et al., 2016*). However, since T cell activation in some cases does not require the TCR (*Beyersdorf et al., 2005*; *Siefken et al., 1998*), we also tested whether the burst in translation controlled by the *TCRA* and *TCRB* mRNA 3'-UTRs could be induced by activation of either the TCR or CD28 individually (*Figure 4*). We found anti-CD28 stimulation alone was sufficient to cause a transient burst in translation in the nanoluciferase reporters with the WT *TCRA* or *TCRB* mRNA 3'-UTRs, whereas anti-CD3 stimulation caused a continuous increase in reporter protein expression (*Figure 4B–D*). Interestingly, the burst in translation required the reporters to be membrane tethered via a N-terminal transmembrane helix (from CD3zeta) that is co-translationally inserted into the membrane (*Call and Wucherpfennig, 2005*; *Figure 4A–D* and *Figure 4—figure supplement 1A–1C*), consistent with the fact that CD28 signaling involves multiple membrane-associated events (*Boomer and Green, 2010*). Moreover, the reporters with *TCRA ΔPAR* and *TCRB ΔPAR* 3'-UTRs failed to show a burst in expression even when the reporter proteins were tethered to the membrane (*Figure 4A–D*), supporting the role for eIF3 in the translational burst. Taken together, these results support the model that the CD28 costimulatory pathway drives an early burst in TCR translation after T cell activation, mediated by eIF3 binding to the *TCRA* and *TCRB* 3'-UTRs.

As noted above, CD28 signaling involves multiple membrane-associated events leading to activation of various kinases, including AKT and mTOR (*Boomer and Green, 2010*; *Mondino and Mueller, 2007*; *Rudd et al., 2009*). We therefore tested whether these kinases are responsible for the signaling downstream of CD28 that leads to eIF3-mediated dynamic regulation of reporter expression, by using inhibitors of the kinases AKT (*Choi et al., 2016*; *Wu et al., 2020*) and mTOR (*Thoreen et al., 2009*). In T cells expressing the membrane-tethered nanoluciferase reporter mRNA with the WT *TCRA* 3'-UTR (*Figure 4A*), treatment with the mTOR inhibitor Torin 1 had no effect on the rapid increase in reporter expression as seen with the DMSO control. However, the burst in nanoluciferase expression was blocked when AKT kinase activity was inhibited with AZD5363 (AZD) (*Figure 4E–F* and *Figure 4—figure supplement 1D–1E*). Altogether, these data indicate that the transient burst in TCR expression likely requires specific interactions between eIF3 and the *TCRA* and *TCRB* 3'-UTRs and also membrane-proximal CD28 signaling. Furthermore, these results support a model in which T cell activation requires CD28 costimulation to elicit an initial positive signal involving AKT kinase activity, which is later repressed by a negative feedback loop also mediated by CD28 signaling and cis-regulatory elements in the *TCRA* and *TCRB* mRNA 3'-UTRs.

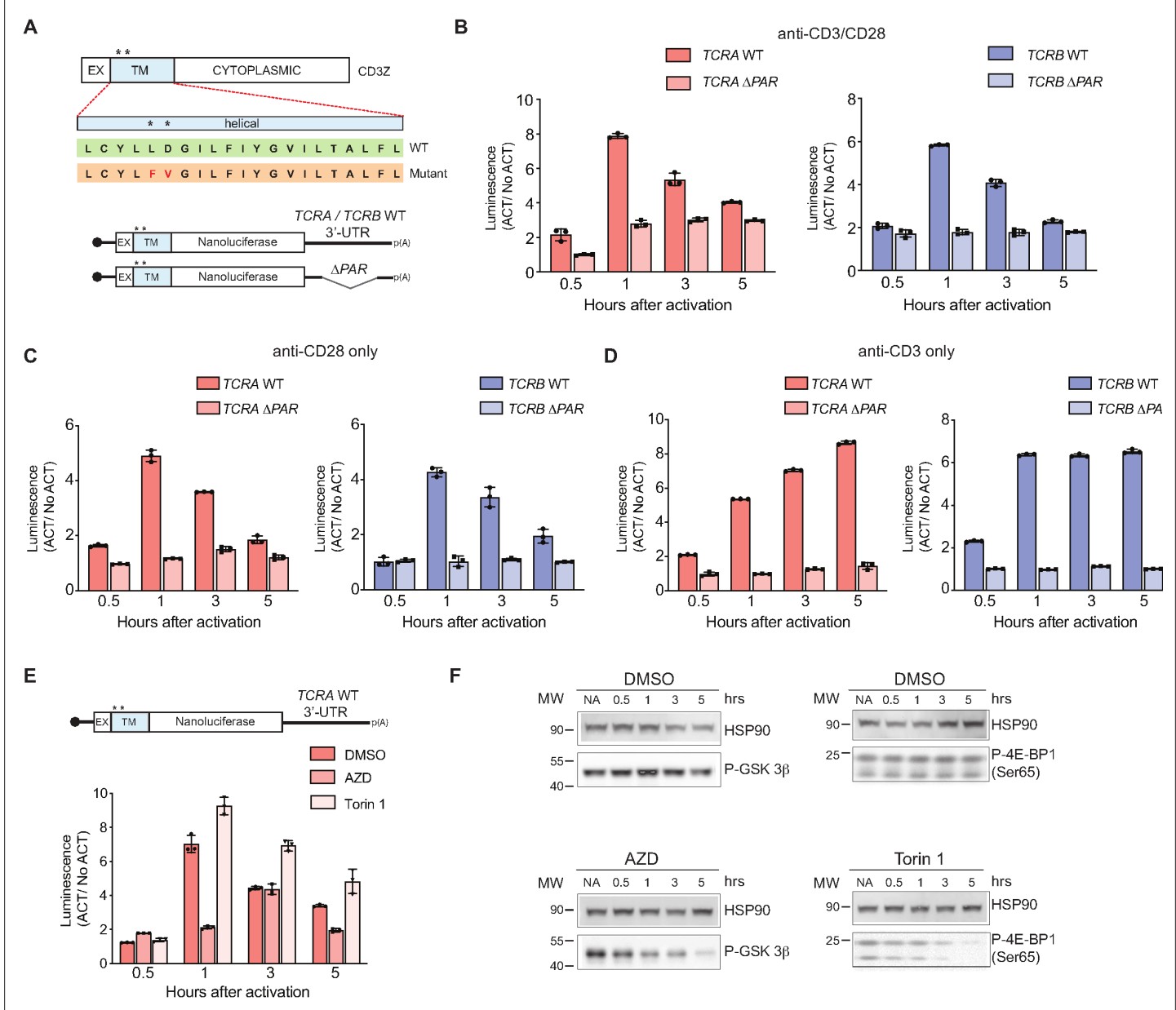

**Figure 4.** The burst in nanoluciferase reporter translation requires membrane-proximal CD28 signaling. (**A**) Schematic of the membrane-tethered nanoluciferase reporters stably expressed in primary human T cells. (Top) Wild-type CD3-zeta protein with asterisk indicating two amino acids mutated in the transmembrane region to prevent association with the TCR (***Dong et al., 2019***) (EX: extracellular, TM: transmembrane). (Bottom) Schematic of the nanoluciferase reporters. Nanoluciferase is fused C-terminal to the extracellular and transmembrane segments of CD3-zeta, mutated to prevent TCR association. The reporters carry the *HBB* 5'-UTR and *TCRA, TCRB, TCRA ΔPAR,* or *TCRB ΔPAR* 3'-UTR. (**B**) Luciferase activity in primary human T cells stably expressing membrane-tethered reporters described in **A** stimulated with anti-CD3/anti-CD28 antibodies, relative to non-activated controls (ACT/NoACT). (**C**) Luciferase activity in primary human T cells stably expressing membrane-tethered reporters described in **A**, and activated only with anti-CD28 antibodies, relative to non-activated controls. (**D**) Luciferase activity in primary human T cells stably expressing membrane-tethered reporters described in **A**, and activated only with anti-CD3 antibodies, relative to non-activated controls. (**E**) Luciferase activity in primary human T cells stably expressing membrane-tethered reporter with WT *TCRA* 3'-UTR described in **A** inhibited with either AZD5363 (AZD) to inhibit AKT activity or Torin 1 to inhibit mTOR before activating with anti-CD3/anti-CD28 antibodies, relative to non-activated controls. In panels A–E, all experiments were carried out in triplicate (three separate wells per condition), with mean and standard deviations shown. (**F**) Western blot carried out to measure AKT activity in the presence of AZD5363 (AZD) using an anti-Phospho-GSK-3β antibody or mTOR activity in the presence of Torin 1 using an anti-Phospho-4EBP1 antibody, for the samples in **E**. HSP90 was used as a loading control. All the primary human T cell experiments were performed using two donors and data from one representative donor is shown.

The online version of this article includes the following figure supplement(s) for figure 4:

**Figure supplement 1.** Nanoluciferase reporter expression in activated T cells.

## eIF3 interactions with the *TCRA* and *TCRB* mRNA 3'-UTRs regulate a burst in TCR translation important for T cell activation

Given that the *TCRA* and *TCRB* mRNA 3'-UTRs are necessary and sufficient to drive a burst in translation after T cell activation, we examined their effects on the endogenous levels of the TCR. We analyzed the expression of TCR protein levels using western blots probed with an anti-TCRA antibody, since the formation of an intact TCR is required to stabilize both the TCRA and TCRB subunits (*Koning et al., 1988*; *Ohashi et al., 1985*). As seen with the nanoluciferase reporters, TCRA levels rose and peaked approximately one hour after T cell activation with anti-CD3/anti-CD28 antibodies (*Figure 5A*). Furthermore, the early burst in TCRA translation is dependent on CD28 but not on TCR signaling (*Figure 5A*), also as observed with membrane-tethered reporters (*Figure 4*). To more directly assess the role of the *TCRA* and *TCRB* mRNA 3'-UTRs, we used CRISPR-Cas9 genome editing to delete the eIF3 PAR-CLIP sites in either the *TCRA* or *TCRB* mRNA 3'-UTRs in primary T cells from two healthy human donors (*Figure 5—figure supplement 1A* and *Supplementary file 6*). PCR analysis showed successful deletion of the eIF3 PAR-CLIP site in the *TCRA* 3'-UTR or in the *TCRB* 3'-UTR (*TCRA ΔPAR* or *TCRB ΔPAR*, respectively) in 43–49% of the alleles (*Figure 5—figure supplement 1B*). A scrambled sgRNA (SC), which does not target any site in the human genome was used as a control. We first measured the total endogenous TCRA protein levels by western blot in *TCRA* or *TCRB* 3'-UTR edited cells versus control cells at different time points after activating with anti-CD3/anti-CD28 antibodies (*Figure 5B*). The SC control cells – which should behave as wild-type T cells – exhibited a substantial burst in TCRA protein levels immediately after activation (~1 hr). By contrast, TCRA protein levels were nearly absent or clearly reduced at early time points after activation in both *TCRA ΔPAR* and *TCRB ΔPAR* cell populations (*Figure 5B*). Only at later time points did TCRA levels in the *TCRA ΔPAR* and *TCRB ΔPAR* cell populations begin to increase. These results share the same pattern of expression seen with the nanoluciferase reporters described above (*Figures 3 and 4*).

We next asked whether the burst in TCR expression driven by the *TCRA* and *TCRB* 3'-UTRs affected downstream steps in T cell activation. During TCR-dependent T cell activation, membrane proteins on the cell surface reorganize at the interface between the T cell and antigen-presenting cell (APC) to form an immunological synapse (*Huppa and Davis, 2003*). TCR cluster formation is a central aspect of immune synapse formation (*Cochran et al., 2001*). Therefore, we tested whether TCR cluster formation is affected by reduced TCR protein levels in *TCRA ΔPAR* and *TCRB ΔPAR* T cells at early time points after activation. We used the same cell populations as those used for the western blots above (*Figure 5B*), to correlate the TCRA protein levels observed in the western blots with TCR clustering. We performed immunofluorescence on SC, *TCRA ΔPAR* and *TCRB ΔPAR* T cell populations using anti-TCRA and anti-TCRB antibodies to detect the TCR. Both *TCRA ΔPAR* and *TCRB ΔPAR* cells had fewer cells forming TCR clusters, especially at the early time point compared to SC control cells when activated with anti-CD3/anti-CD28 (*Figure 5C* and *Figure 5—figure supplement 1C*). Consistent with the TCRA expression levels observed in western blots and the rate of TCR clustering (*Figure 5B and C*), both *TCRA ΔPAR* and *TCRB ΔPAR* cells expressed lower amounts of cell surface TCR compared to SC control cells or cells edited with a single gRNA when tested by flow cytometric analysis after PMA + I activation (*Figure 5—figure supplement 1D and E*).

To test whether the defect in TCR clustering in the *TCRA ΔPAR* and *TCRB ΔPAR* cell populations reflects a general deficiency in T cell activation, we measured the T cell activation markers CD69 and CD25 (subunit IL2RA of the IL2 receptor) on the cell surface by flow cytometry (*Figure 5—figure supplement 1F* and *Figure 5—figure supplements 2–4* ). Fewer cells in the *TCRA ΔPAR* and *TCRB ΔPAR* CD8+ and CD4+ primary T cell populations expressed CD69 at early time points after activation (5–8 hr) (*Figure 5—figure supplement 4A and B*) and fewer expressed both CD69 and CD25 at later time points after activation, compared to SC control cells (*Figure 5D–E* and *Figure 5—figure supplement 4C and D*). We also found the *TCRA ΔPAR* and *TCRB ΔPAR* T cell populations secreted lower amounts of the stimulatory cytokines IL2 (*Figure 5F* and *Figure 5—figure supplement 4E*) and IFNγ (*Figure 5G* and *Figure 5—figure supplement 4F*), compared to the SC control cells. Taken together, the *TCRA ΔPAR* and *TCRB ΔPAR* primary T cell populations exhibited multiple early and late T cell activation defects. These results support the model that after T cell activation, eIF3 binding to the *TCRA* and *TCRB* mRNA 3'-UTRs drives an early burst in TCR translation that is required for many subsequent steps in T cell activation.

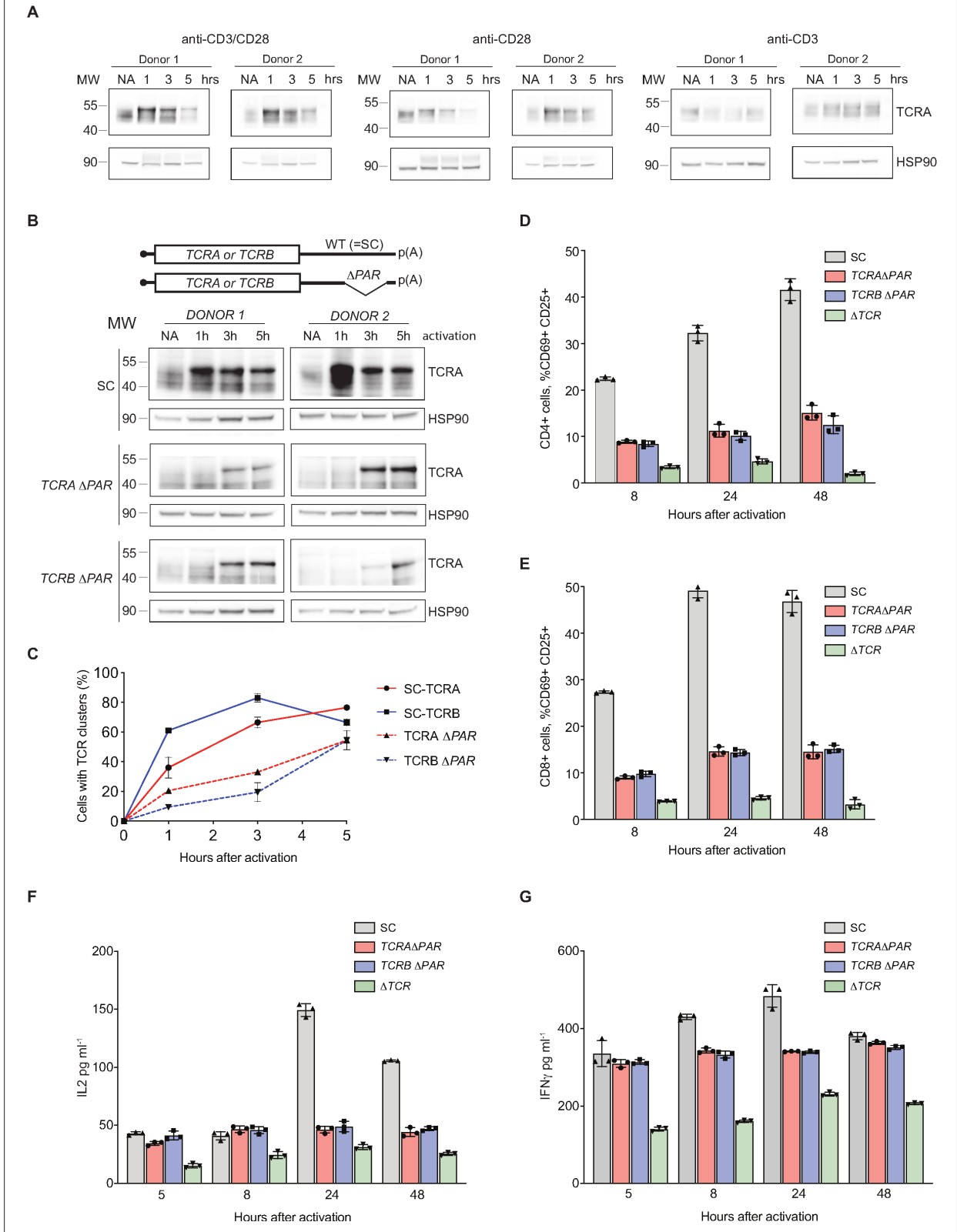

**Figure 5.** eIF3 binding to the *TCRA* and *TCRB* mRNA 3'-UTR elements is required for a rapid burst in TCR translation and robust activation of primary human T cells. (**A**) Western blots of TCRA protein levels in T cells as a function of time after different modes of activation. HSP90 was used as a loading control. (**B**) Western blots measuring TCRA protein levels as a function of time after anti-CD3/anti-CD28 activation. Cell lines used are labeled on the left: *TCRA ΔPAR*, *TCRB ΔPAR*, and SC (scrambled gRNA). HSP90 was used as a loading control. Schematics of *TCRA* and *TCRB* mRNAs with and without

*Figure 5 continued on next page*

*Figure 5 continued*

eIF3 PAR-CLIP sites are shown above. SC control cells have the WT 3'-UTRs for *TCRA* and *TCRB* mRNAs. (**C**) The number of T cells with one or more TCR clusters measured by anti-TCRA/anti-TCRB protein staining and epifluorescence microscopy as a function of time after anti-CD3/anti-CD28 activation. A total of 100 cells from each donor were imaged for *TCRA ΔPAR* (n = 2 donors, stained with anti-TCRA antibody), *TCRB ΔPAR* (n = 2 donors, stained with anti-TCRB antibody), and SC cell lines (n = 2 donors, each stained separately with anti-TCRA and anti-TCRB antibodies). Values are mean ± standard deviation. (**D**) Flow cytometric analysis measuring T cell activation markers CD69 (early activation marker) and CD25 (mid-activation marker), quantifying the mean percent of CD4+ T cells that are CD69+ CD25+. (**E**) Flow cytometric analysis of CD8+ T cells, quantifying the mean percent of CD8+ T cells that are CD69+ CD25+. Cells sorted as shown in *Figure 5—figure supplement 2*. (**F**) Quantification of IL2 secreted from SC, *TCRA ΔPAR*, *TCRB ΔPAR*, and *ΔTCR* cell populations at different time points after stimulation with anti-CD3/anti-CD28 antibodies, as determined by ELISA. (**G**) Quantification IFNγ secreted from the cells in **F**, as determined by ELISA. In panels **D–G**, all experiments were carried out in triplicate (three separate wells per condition), with mean and standard deviations shown. Representative results from one donor are shown.

The online version of this article includes the following figure supplement(s) for figure 5:

**Figure supplement 1.** Generation and analysis of *TCRA ΔPAR* and *TCRB ΔPAR* primary human T cells.

**Figure supplement 2.** Effects of *TCRA ΔPAR* and *TCRB ΔPAR* mutations on different steps of CD8+ T cell activation.

**Figure supplement 3.** Effects of *TCRA ΔPAR* and *TCRB ΔPAR* mutations on different steps of CD4+ T cell activation.

**Figure supplement 4.** Effects of *TCRA ΔPAR* and *TCRB ΔPAR* mutations on different steps of T cell activation.

**Figure supplement 5.** eIF3 binding to the *TCRA* and *TCRB* mRNA 3'-UTR elements in Jurkat cells.

To obtain additional mechanistic understanding of eIF3-mediated regulation of TCR translation we generated clonal *TCRA ΔPAR* and *TCRB ΔPAR* Jurkat cells using the CRISPR-Cas9 genome editing strategy developed for primary T cells (*Figure 5—figure supplement 1A*, *Supplementary file 6*). We first measured total TCR levels using western blots and an anti-TCRA antibody as described above (*Figure 5A*), at different time points after activation with anti-CD3/anti-CD28 antibodies. WT Jurkat cells showed a TCR translational burst that peaked 5–8 hr after activation (*Figure 5—figure supplement 5A*). By contrast, both *TCRA ΔPAR* and *TCRB ΔPAR* Jurkat cell populations expressed lower levels of the TCR proteins compared to WT cells, and failed to show a burst in TCR expression at early time points after activation (*Figure 5—figure supplement 5A*). Importantly, *TCRA* and *TCRB* mRNA levels were unaffected or even increased in the *TCRA ΔPAR* and *TCRB ΔPAR* Jurkat cells (*Figure 5—figure supplement 5B and C*), similar to our observations with nanoluciferase reporters in primary T cells (*Figure 3C*). This is consistent with TCR expression levels being regulated post-transcriptionally by eIF3 interactions with the *TCRA* or *TCRB* mRNA 3'-UTR elements. We then tested whether deleting eIF3 binding sites in both *TCRA* and *TCRB* mRNA 3'-UTRs affected their interaction with eIF3. In both *TCRA ΔPAR* and *TCRB ΔPAR* cells, eIF3 bound to significantly lower amounts of the *TCRA* and *TCRB* mRNAs compared to WT cells at both the 5 hr and 8 hr time points after anti-CD3/anti-CD28 activation (*Figure 5—figure supplement 5D and E*). This indicates deleting the eIF3 PAR-CLIP sites in the 3'-UTR disrupts eIF3 interactions with the *TCRA* and *TCRB* mRNAs substantially. Together with the nanoluciferase reporter experiments in primary T cells (*Figure 3*), these results support the model that eIF3 binding to the 3'-UTR elements of the *TCRA* and *TCRB* mRNAs mediates the rapid burst in TCR translation after T cell activation.

## The *TCRA* and *TCRB* mRNA 3'-UTRs enhance anti-CD19 CAR T cell function

T cells engineered to express chimeric antigen receptors (CARs) for cancer immunotherapy now in use clinically employ artificial 3'-UTRs in the CAR-encoding mRNA, a woodchuck hepatitis viral post-transcriptional regulatory element (*WPRE*) (*Milone et al., 2009*), or a retroviral 3'-long terminal repeat (*3'-LTR*) (*Kochenderfer et al., 2009*). However, it is not known whether these 3'-UTR elements provide optimal CAR expression or CAR T cell function. To test whether these 3'-UTRs induce a transient burst in translation as observed with the WT *TCRA* or *TCRB* 3'-UTRs, we fused the *WPRE* and *3'-LTR* 3'-UTR sequences to nanoluciferase reporters and expressed these in primary T cells (*Figure 6A*). In contrast to the *TCRA* and *TCRB* 3'-UTRs (*Figures 3 and 4*), the *WPRE* and *3'-LTR* 3'-UTR elements failed to induce the early burst in nanoluciferase expression (*Figure 6A*). These data suggest that fusing the *TCRA* or *TCRB* 3'-UTR sequences to engineered CAR open reading frames could be used to obtain more physiological expression dynamics seen for the endogenous TCR.

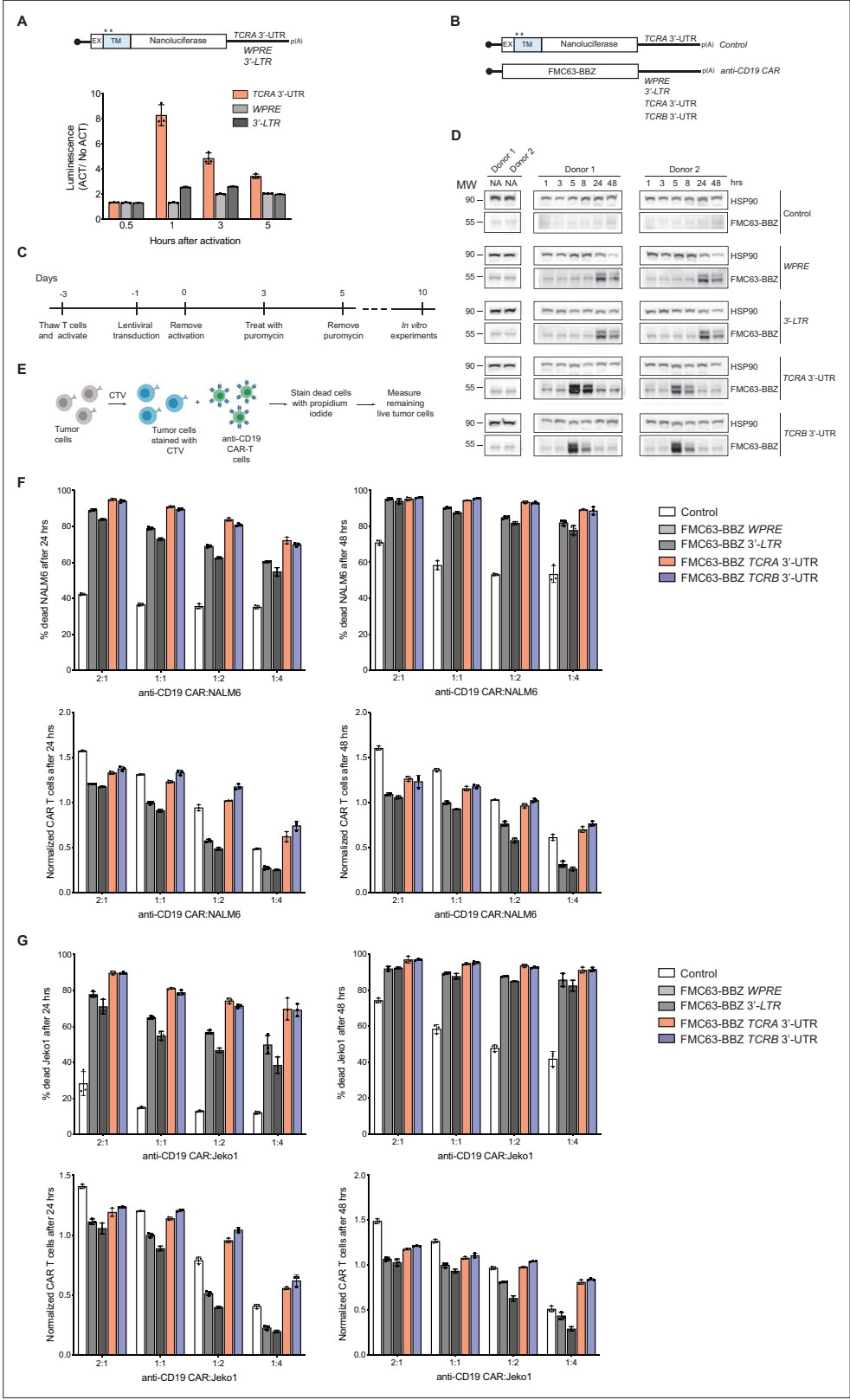

**Figure 6.** The *TCRA* and *TCRB* mRNA 3'-UTR elements enhance CAR T cell function. (**A**) (Top) Schematic of membrane-tethered nanoluciferase reporters stably expressed in primary human T cells. The reporters have the *HBB* 5'-UTR and either the *TCRA* 3'-UTR, the Woodchuck Hepatitis Virus Posttranscriptional Regulatory Element (*WPRE*) or the gammaretroviral 3'-Long Terminal Repeat (*3'-LTR*) as 3'-UTR. (Bottom) Luciferase activity in primary

*Figure 6 continued on next page*

*Figure 6 continued*

human T cells stably expressing membrane-tethered nanoluciferase reporters described above and activated with anti-CD3/anti-CD28 antibodies, relative to non-activated cells. Representative results from one of two donors are shown (n = 3 separate wells, with mean and standard deviations shown). (**B**) Schematics of FMC63-BBZ CAR cDNA sequence fused 5′ of the *WPRE, 3′-LTR, TCRA* 3′-UTR or *TCRB* 3′-UTR. The membrane-tethered nanoluciferase reporter with *TCRA* 3′-UTR was used as a control for the effects of lentiviral expression. (**C**) Timeline of CAR T cell generation from primary human T cells and the experiments performed. (**D**) Western blots measuring FMC63-BBZ CAR protein levels as a function of time after incubation with NALM6 cells. CAR T cells expressing the constructs in **B** are labeled: Control, *WPRE, 3′-LTR, TCRA* and *TCRB* 3′-UTR. HSP90 was used as a loading control (n = 2 donors). (**E**) Schematic describing the cytotoxicity assay used to detect live tumor cells after incubation with FMC63-BBZ CAR cells using flow cytometric analysis. (**F**) Top panels show cytotoxic activity of FMC63-BBZ CARs fused to various 3′-UTRs described in **B** after incubating with NALM6 cells for 24 and 48 hr. Bottom panels show CAR T cells present in the experiments at the different time points, normalized to the *WPRE* CAR T cells at a 1:1 ratio with target tumor cells. (**G**) Top panels show cytotoxic activity of FMC63-BBZ CARs fused to various 3′-UTRs described in **B** after incubating with Jeko 1 cells for 24 and 48 hr. Bottom panels show CAR T cells present in the experiments at the different time points, normalized to the *WPRE* CAR T cells at a 1:1 ratio with target tumor cells. In panels **F** and **G**, representative results for one of two donors are shown for T cells propagated in 50 U/mL IL2 (n = 3 separate wells, with mean and standard deviations shown).

The online version of this article includes the following figure supplement(s) for figure 6:

**Figure supplement 1.** Effects of the *TCRA* and *TCRB* mRNA 3′-UTR elements on CAR T cell function.

**Figure supplement 2.** Effects of the *TCRA* and *TCRB* mRNA 3′-UTR elements on CAR T cell function, donor 2.

**Figure supplement 3.** Anti-CD19 CAR T cell killing assays for T cells propagated in 500 U/mL IL2, donor 1.

**Figure supplement 4.** Anti-CD19 CAR T cell killing assays for T cells propagated in 500 U/mL IL2, donor 2.

**Figure supplement 5.** Effects of the *TCRA* and *TCRB* mRNA 3′-UTR elements on CAR T cell function, in cells with TCR expression knocked out.

**Figure supplement 6.** Effects of the *TCRA* and *TCRB* mRNA 3′-UTR elements on CAR T cell function for T cells expressing an FMC63-28Z CAR.

We therefore engineered primary human T cells to express an anti-CD19 CAR currently in use clinically to treat B cell lymphomas (*Kalos et al., 2011*; *Milone et al., 2009*; *June et al., 2014*). We used lentiviral transduction to express the anti-CD19 CAR from mRNAs with either *WPRE, 3′-LTR, TCRA* or *TCRB* 3′-UTRs to make anti-CD19 CAR T cells (*Figure 6B and C*). We then stimulated these CAR T cells by incubating them with CD19-expressing leukemia cells (NALM6) and measured the CAR expression by western blot analysis at different time points. Interestingly, the *TCRA* and *TCRB* 3′-UTRs induced a burst in anti-CD19 CAR protein levels within 5 hr of NALM6 cell addition, whereas the *WPRE* and *3′-LTR* sequences delayed the burst in CAR expression to ~24 hr (*Figure 6D*). In these CAR T cells, TCR expression also followed the burst in CAR expression, and was dependent on the presence of the CAR in addition to the NALM6 cells (*Figure 6—figure supplement 1A*).

Lastly, we tested whether the timing of CAR protein expression correlates with the killing capacity of these CAR T cells. Using in vitro cytotoxicity assays (*Figure 6E*), CAR T cells expressing the CAR using either the *TCRA* or *TCRB* 3′-UTR showed more robust killing of two tumor cell lines compared to CARs using either the *WPRE* or *3′-LTR* 3′-UTRs. This was true for CAR T cells propagated using two different concentrations of IL2 in the media (50 U/mL or 500 U/mL) (*Figure 6F–G*, *Figure 6—figure supplement 1B–1D*, *Figure 6—figure supplement 2*, and *Figure 6—figure supplements 3 and 4*). Interestingly, CAR T cells expressing the CAR using either the *TCRA* or *TCRB* 3′-UTR generally propagated more robustly in the presence of the target tumor cells regardless of condition (*Figure 6F–G*, *Figure 6—figure supplement 1B–1D*, *Figure 6—figure supplement 2*, and *Figure 6—figure supplements 3 and 4*). We also tested CAR T cells in which TCR expression was knocked out (*Figure 6—figure supplement 5A and B*). As observed above, these CAR T cells expressing the CAR using either the *TCRA* or *TCRB* 3′-UTR also showed more robust killing and propagated more robustly in the presence of target tumor cells (*Figure 6—figure supplement 5C and D*).

We also tested a separate anti-CD19 CAR used in the clinic that uses a CD28-derived signaling domain (*Figure 6—figure supplement 6A*; *Kochenderfer et al., 2010*; *Kochenderfer et al., 2009*) rather than the 4-1BB derived signaling domain tested above (*Figure 6B*; *Kalos et al., 2011*; *Milone et al., 2009*; *June et al., 2014*). As observed for the 4-1BB domain-containing CAR, the CD28

domain-containing CAR also exhibited more potent killing and CAR T cells propagated more robustly in the presence of target tumor cells when the CAR expression was driven by the *TCRA* or *TCRB* 3'-UTRs (*Figure 6—figure supplement 6B and C*). Taken together, these results are consistent with the importance of an early burst of CAR translation for optimal CAR T cell function. These results also support the idea that using the native *TCRA* or *TCRB* 3'-UTRs for CAR expression can be used to improve CAR T cell function.

## Discussion

The eIF3 PAR-CLIP experiment we present here provides a snapshot of eIF3-RNA interactions that occur at the time TCR translation is most sensitive to eIF3 regulation (5 hr in Jurkat cells, *Figure 5—figure supplement 3A*). At this point in time, eIF3 crosslinks to multiple mRNAs encoding proteins involved in immune cell function (*Supplementary files 2 and 3*). Interestingly, the patterns of eIF3 crosslinking, which for a number of mRNAs include interactions with the protein coding sequence and 3'-UTR (*Figure 1* and *Figure 1—figure supplement 1G and H*), suggest an active role for eIF3 in promoting translation of these mRNAs. In support of this model, the two examples of pan-mRNAs we examined here (*TCRA* and *TCRB*) do not colocalize with P bodies or stress granules (*Figure 1F*) and binding to eIF3 includes the mRNA coding sequence on translating ribosomes (*Figure 2*). However, this eIF3-mediated translation activation is transient, lasting only 1–2 hr in primary T cells (*Figure 5A*). Importantly, the *TCRA* and *TCRB* mRNA 3'-UTR elements are necessary and sufficient to control this burst in translation (*Figure 3*). Recent evidence suggests that eIF3 can remain associated with translating ribosomes (*Bohlen et al., 2020*; *Lin et al., 2020*; *Wagner et al., 2020*), a phenomenon that seems to be enhanced for the pan-mRNAs identified here. It is possible that the 5'-UTRs and 3'-UTRs of some mRNAs are in close proximity, that is on the membrane of the endoplasmic reticulum (*Christensen et al., 1987*; *Christensen and Bourne, 1999*), and eIF3 binding to the 3'-UTR could directly influence the efficiency of translation initiation from the 5'-UTR. Additional layers of translation regulation also contribute to T cell function (*Tan et al., 2017*), particularly with respect to mTOR signaling (*Miyamoto et al., 2005*; *Myers et al., 2019*) and carbon metabolism (*Manfrini et al., 2017*; *Ricciardi et al., 2018*). The present PAR-CLIP experiments should help to elucidate additional roles for eIF3-mediated translation regulation and to map the system-wide role of translation regulation in T cell activation.

Recent experiments indicate that T cells must cross a threshold of T cell receptor signaling to commit to activation and proliferation (*Au-Yeung et al., 2017*; *Au-Yeung et al., 2014*), setting up a 'digital' response to antigen recognition (*Allison et al., 2016*; *Au-Yeung et al., 2017*; *Au-Yeung et al., 2014*; *Richard et al., 2018*). The response threshold involves integration of intensity and duration of TCR signaling (*Au-Yeung et al., 2017*; *Au-Yeung et al., 2014*; *Richard et al., 2018*), and spans a wide range of TCR antigen affinity (*Allison et al., 2016*; *Au-Yeung et al., 2017*; *Au-Yeung et al., 2014*; *Richard et al., 2018*). Notably, T cell commitment to clonal expansion and differentiation can occur within as little as 1–2 hr of TCR stimulation for effector CD4+ and naive CD8+ T cells (*Iezzi et al., 1998*; *van Stipdonk et al., 2001*). Remarkably, this time period spans the burst in TCR protein synthesis mediated by eIF3 interactions with the *TCRA* and *TCRB* mRNA 3'-UTR elements (*Figure 5A*). Naive CD4+ T cells require a longer duration of TCR signaling of ~20 hr (*Iezzi et al., 1998*; *Schrum et al., 2005*). Our observation of the burst in TCRA translation (and by implication TCRB translation) (*Koning et al., 1988*; *Ohashi et al., 1985*) in the first 1–2 hr of stimulation also correlates with multiple downstream events required for T cell activation and function. For example, loss of the burst in TCR protein synthesis delayed formation of TCR clusters (*Figure 5C*), suggesting that surface expression of the TCR (*Cochran et al., 2001*; *Huppa and Davis, 2003*) is also impacted by the dynamics of TCRA and TCRB translation. Although we were not able to distinguish levels of TCR translation in isolated CD8+ and CD4+ T cells (*Figure 5*), subsequent events in T cell activation including CD69 and CD25 expression, and IL2 and IFNγ secretion, were equally affected in CD8+ and CD4+ cells in which the *TCRA* or *TCRB* 3'-UTR PAR-CLIP sites were deleted (*Figure 5C–F*). In an immune response, CD28 engagement serves as the second signal required for T cell activation (*Harding et al., 1992*; *Harding and Allison, 1993*) and affects the first minutes of TCR-mediated signaling (*Green et al., 2000*; *Green et al., 1994*; *Michel et al., 2001*; *Shahinian et al., 1993*; *Tuosto and Acuto, 1998*). Here, we show CD28-mediated signaling is also needed for the burst of TCR translation on the hour timescale in primary T cells (*Buckler et al., 2006*; *Figure 4*). Taken together, our results indicate that eIF3 controls

*TCRA* and *TCRB* mRNA translation during the first critical hours after antigen recognition that leads to subsequent T cell commitment to proliferation and differentiation (*Figure 5*).

We found the CD28-dependent burst in TCR translation requires cotranslational membrane targeting, consistent with the fact that CD28-mediated signaling occurs at the plasma membrane (*Boomer and Green, 2010*) and the fact that both the TCRA and TCRB subunits are integral membrane proteins. We also observed additional immune-related mRNAs in the PAR-CLIP experiment that encode membrane proteins or secreted proteins (i.e. B2M, CD28, CD3D, HLA-E, and LAT in *Supplementary file 5*, and TAP1 and TAP2 in *Supplementary file 3*) which may be particularly sensitive to CD28 signaling. The requirement for CD28 in PD-1 mediated inhibition of T cell activation (*Hui et al., 2017*; *Kamphorst et al., 2017*) also suggests eIF3-mediated control of TCR expression may affect PD-1 checkpoint blockade-based cancer immunotherapy (*Jiang et al., 2019*). It will be interesting to determine if there is a connection between PD1-mediated inhibition of the T-cell response and eIF3 binding to the 3'-UTRs of the *TCRA* and *TCRB* mRNAs. It will also be important to map the CD28-downstream signaling pathway to understand both the increase and subsequent decrease in TCR translation that occur during the translation burst. We found that AKT influences the rapid rise in translation (*Figure 4E–F*), but other kinases such as Tec-kinases (*Andreotti et al., 2018*; *Gallagher et al., 2021*; *Hallumi et al., 2021*) could also impact the translation burst.

Cell immunotherapies targeting various cancers have made great strides, due to the engineering of chimeric antigen receptors that couple antigen recognition to intracellular signaling domains that activate cytotoxic T cells. However, CAR T cells often fail to eradicate cancers due to loss of activity over time, for example T cell exhaustion (*Globerson Levin et al., 2021*; *Watanabe et al., 2018*). Our results using nanoluciferase reporters indicated that eIF3-responsive mRNA 3'-UTR elements could be used to improve chimeric antigen receptor expression and CAR T cell responsiveness (*Eyquem et al., 2017*; *Globerson Levin et al., 2021*; *Watanabe et al., 2018*). We used this information to improve the ability of CAR T cells to kill tumor cells in vitro. With a clinically validated anti-CD19 CAR, we found that using the *TCRA* or *TCRB* mRNA 3'-UTRs dramatically shortened the lag time before the burst in CAR protein expression upon exposure to tumor cells (*Figure 6D*). This short lag time correlated with improved tumor cell killing in cytotoxicity assays (*Figure 6F and G*, *Figure 6—figure supplements 1–6*). Interestingly, using the *TCRA* or *TCRB* mRNA 3'-UTRs for CAR expression also improved CAR T cell propagation in the presence of target tumor cells (*Figure 6F and G*, *Figure 6—figure supplements 1–6*). Future experiments will be needed to determine the physiological basis for improved propagation of these CAR T cells in the presence of target tumor cells, and to address whether the timing of the CAR translational burst affects CAR T cell exhaustion.

There are fundamental differences between TCRs and CARs used in the clinic that may not be addressed using the modular nature of the *TCRA* and *TCRB* 3'-UTRs. For example, CARs generally have a weaker affinity for their target ligand compared to TCRs for antigens presented on the cell surface (*Watanabe et al., 2018*). Furthermore, CARs do not elicit immune synapses as observed for TCRs (*Dong et al., 2020*). Here, we observed the burst in CAR protein expression using the *TCRA* or *TCRB* 3'-UTRs was still not as rapid as that of the endogenous TCR in T cells stimulated by anti-CD3/anti-CD28 antibodies (*Figure 5A*). This could be due to multiple factors, including CAR affinity for the CD19 antigen or the activity of the 4-1BB costimulatory domain used in the present CAR (*Kalos et al., 2011*; *Milone et al., 2009*; *June et al., 2014*) instead of the CD28 signaling domain (*Kochenderfer et al., 2009*). It is also possible that the 3'-UTRs of the endogenous TCR subunits, which were left intact in most of the present cytotoxicity assays, and which also responded to CAR signaling (*Figure 6—figure supplement 1A*), titrated cellular factors required for even shorter response times. T cell engineering to knock out the endogenous TCR 3'-UTRs or improved CAR design may shorten the lag before the burst in CAR expression and further improve CAR T cell function. Taken together, our experiments delineate the central role of eIF3 in T cell activation and highlight the importance of understanding translation regulation in immune cells to open new avenues for engineering improved cell therapies (*Figure 7*).

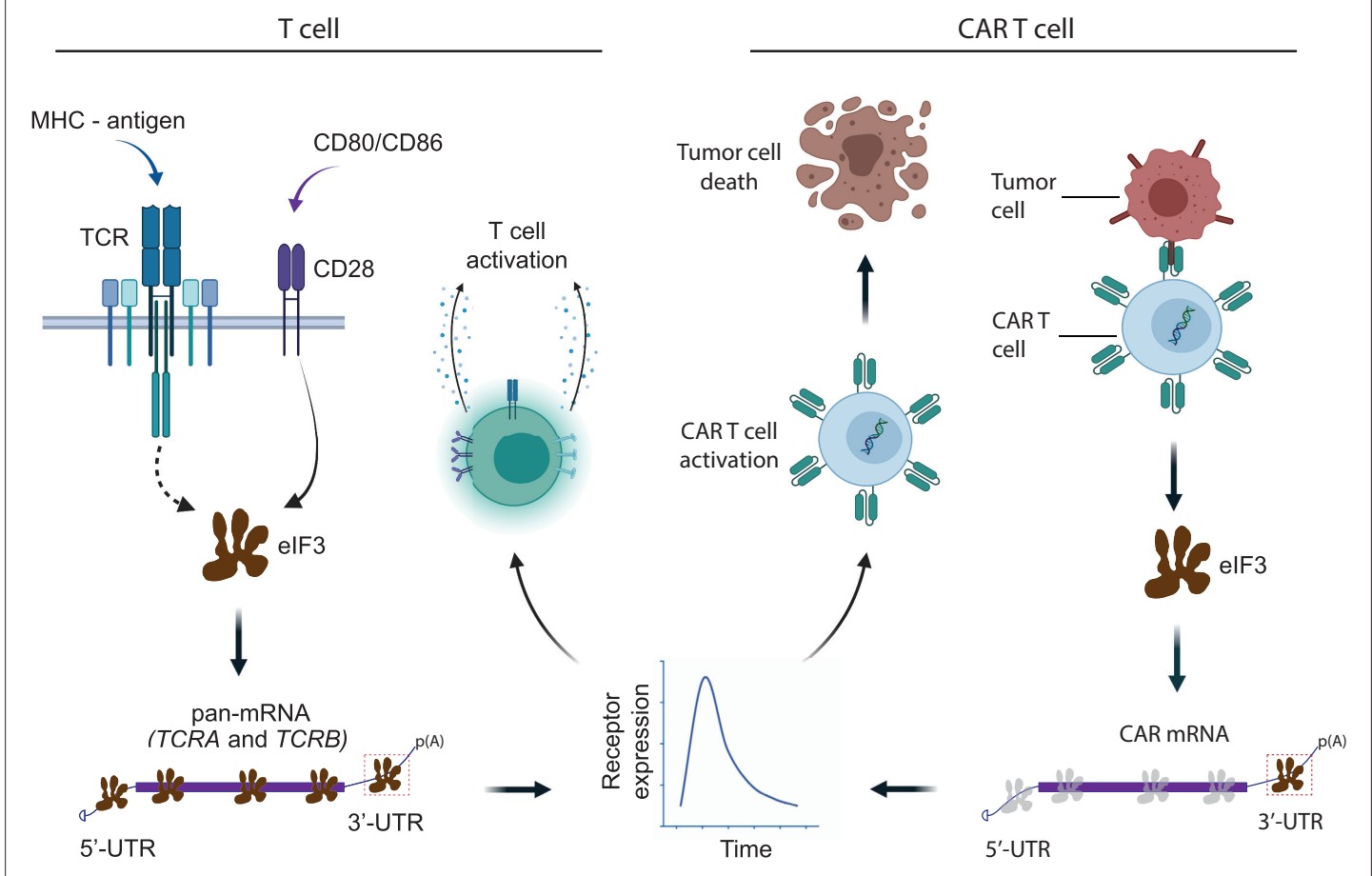

**Figure 7.** Model for robust T cell activation and improved CAR T cell function mediated by eIF3 interactions with the *TCRA* and *TCRB* mRNAs. Left, a T cell activated upon antigen recognition by the TCR and CD28 costimulatory signal leads to recruitment of the *TCRA* and *TCRB* mRNAs to translating ribosomes due to eIF3 binding to the mRNA 3'-UTRs. This results in a short burst in TCRA and TCRB translation followed by a robust increase in T cell function as measured by CD69 and CD25 expression and cytokine release. Right, activation of CAR T cells expressing CAR mRNAs with the *TCRA* or *TCRB* 3'-UTRs leads to a burst in CAR translation and improved tumor cell killing.

## Materials and methods

### Jurkat cell culture

Human Jurkat Clone E6-1 (ATCC TIB-152) was purchased from American Type Culture Collection (ATCC). Cells were routinely tested for mycoplasma contamination at the UC Berkeley Cell Culture Facility. Cells were maintained in RPMI 1640 Medium (ATCC modification) with 10% FBS (VWR Life Science Seradigm) and 0.01% Penicillin-Streptomycin (10,000 U/mL) (ThermoFisher). The cells were maintained between $1 \times 10^5$ cells ml$^{-1}$ to $8 \times 10^5$ cells ml$^{-1}$. When cells were stimulated, they were always maintained at $8 \times 10^5$ cells ml$^{-1}$.

### Jurkat cell stimulation

The Jurkat cells used for the PAR-CLIP experiment were stimulated with 1 X Cell Stimulation Cocktail, containing phorbol 12-myristate 13-acetate (PMA) and ionomycin (ThermoFisher, Cat. #: 00-4970-93) to ensure a large proportion of the cells were activated. Unless otherwise stated, all other experiments involving activated Jurkat cells used anti-CD3/anti-CD28 antibodies (Tonbo) for stimulation. Flat bottom plates were coated with anti-CD3 antibody at a 10 µg/mL concentration, and anti-CD28 antibody was added to the cell culture media at a concentration of 5 µg/mL.

## Isolation of human primary T cells

Primary human T cells were isolated from healthy human donors from leukoreduction chambers after Trima Apheresis (Vitalant, formerly Blood Centers of the Pacific). Peripheral blood mononuclear cells (PBMCs) were isolated from whole blood samples by Ficoll centrifugation using SepMate tubes (STEMCELL) per manufacturer's instructions. T cells were isolated from PBMCs from all cell sources by magnetic negative selection using an EasySep Human T Cell Isolation Kit (STEMCELL), per manufacturer's instructions. Unless otherwise noted, isolated T cells were stimulated with anti-CD3/anti-CD28 antibodies (Tonbo) as described above and used immediately.

## Primary human T cell culture

Bulk T cells were cultured in XVivo15 medium (Lonza) with 5% fetal bovine serum (FBS), 50 μM 2-mercaptoethanol (Sigma), and 10 μM *N*-acetyl-cystine (Sigma) or ImmunoCult-XF T Cell Expansion Medium (StemCell). Immediately after isolation, T cells were either frozen or stimulated for 2 days with anti-human CD3/CD28 magnetic Dynabeads (ThermoFisher) at a bead to cell concentration of 1:1, along with cytokine IL-2 (UCSF Pharmacy). For T cells cultured after electroporation, the media was supplemented with IL-2 at 500 U/mL. Throughout the culture period T cells were maintained at an approximate density of 1 million cells per mL of media. Every 2–3 days after electroporation, additional media was added, along with fresh IL-2 at 500 U/mL concentration, and the cells were transferred to larger culture flasks as necessary to maintain a density of 1 million cells per mL. For all the other times T cells were maintained at 50 U/mL IL-2 with the addition of fresh media every 2 days.

## Primary human T cell stimulation

For edited primary human T cells, the cells were transferred to fresh media lacking IL-2 after 9 days of culturing. The T cells were then stimulated with anti-CD3/anti-CD28 antibodies using flat bottom plates coated with anti-CD3 antibody (Tonbo) at a 10 μg/mL concentration, and anti-CD28 antibody (Tonbo) added to the cell culture media at a concentration of 5 μg/mL. In all other primary human T cell stimulation experiments the cells were stimulated with anti-CD3/anti-CD28 antibodies as mentioned above along with 50 U/mL IL-2.

## 4-Thiouridine optimization for PAR-CLIP experiments

We used Jurkat cells as a model for T cells, as PAR-CLIP experiments require a large number of cells labeled with 4-thiouridine (Sigma) at a non-toxic concentration (*Ascano et al., 2012*). Jurkat cells also have a defined T cell receptor and transcriptome, avoiding the donor-to-donor variability of primary T cells. Jurkat cells were seeded, so that they reached $8 \times 10^5$ cells ml$^{-1}$ seeding density on the day of the experiment. Varying concentrations of 4-thiouridine (s4U) were added to the cells (50 μM, 75 μM, 100 μM, or none as a negative control). The cells were then incubated for different time points: 8, 10, 12, or 16 hr. After each incubation time, cell viability was determined using the CellTiter-Glo assay (Promega), according to the manufacturer's instructions. Concentrations at which the relative luminescence in the presence of s$^4$U (luminescence of the s$^4$U treated cells/luminescence of the untreated cells) exceeded 95% were considered non-toxic. Based on these measurements, we used 50 μM of 4-thiouridine for PAR-CLIP experiments.

## PAR-CLIP

Two biological replicates were used to perform PAR-CLIP analysis as described in *Lee et al., 2015*, with modifications for Jurkat cells. 50 μM of 4-thiouridine was determined as non-toxic to Jurkat cells over the time course of the PAR-CLIP experiments (*Ascano et al., 2012*). A total of 55 million Jurkat cells seeded at $8 \times 10^5$ cells ml$^{-1}$ was treated with 50 μM of 4-thiouridine for 7 hr, then stimulated with 1 X Cell Stimulation Cocktail for 5 hr in media containing 50 μM of 4-thiouridine (*Figure 1A*). The same number of cells were treated with 50 μM of 4-thiouridine for 12 hr without stimulation as a non-activated control. The cells were then crosslinked on ice with 365 nm UV irradiation at an energy dose of 0.2 J cm$^{-2}$. The cells were pelleted by centrifugation at 100 x g for 15 min at 4 °C, and the pellet was resuspended in three volumes of NP40 lysis buffer (50 mM HEPES-KOH pH 7.5, 150 mM KCl, 2 mM EDTA, 0.5% Nonidet P-40 alternative, 0.5 mM dithiothreitol (DTT), 1 Complete Mini EDTA-free Protease Inhibitor Cocktail tablet (Roche)). The cell suspension was then incubated on ice for 10 min, passed through an 18 G needle five times, and centrifuged at 13,000 x g for 15 min at 4 °C and RNAs

were lightly digested by treatment with MNase (Thermo Scientific) at a final concentration of 0.05 U μl⁻¹ for 20 min at 16 °C. For each PAR-CLIP assay, 1000 μl of Dynabeads (Invitrogen) and 800 μl of anti-EIF3B antibody (Bethyl A301-761A) were used. The remaining steps of the PAR-CLIP analysis were performed exactly as described in *Danan et al., 2016*; *Lee et al., 2015* with the exception of using MNase at 5 U μl⁻¹ for the on-bead digestion step.

## Mass spectrometry

To identify eIF3 subunits that crosslinked with RNAs in the PAR-CLIP experiments, a portion of eIF3 immunoprecipitated using Dynabeads as described above were treated with nonradioactive ATP (NEB) during the T4 polynucleotide kinase labeling step. The nonradioactive samples were then run on the same gel next to the radiolabeled PAR-CLIP samples, Coomassie stained (Pierce) and the bands that matched with the phosphorimager printout were excised from the gel and submitted for identification using one-dimensional LC-MS/MS (*Supplementary file 1*).

## PAR-CLIP computational analysis

PAR-CLIP cDNA libraries were sequenced on an Illumina HiSeq 2,500 platform. To eliminate potential PCR biases during PAR-CLIP library preparation, a random bar code was introduced into the 3′ adapter and all the reads that matched the random barcode were collapsed into single reads. Clusters of overlapping sequence reads mapped against the human genome version hg38 were generated using the PARalyzer software (*Corcoran et al., 2011*) incorporated into the PARpipe pipeline (https://ohlerlab.mdc-berlin.de/software/PARpipe_119/, *Mukherjee et al., 2019*) with the settings below. Binding sites were categorized using the Gencode GRCh38.p12 GTF annotations (gencode.v21.annotation.gtf), https://www.gencodegenes.org/human/.

The PARpipe settings used were:

Conversion = T > C
Minimum read count per group = 5
Minimum read count per cluster = 7
Minimum read count for kde = 3
Minimum cluster size = 11
Minimum conversion locations for cluster = 2
Minimum conversion count for cluster = 2
Minimum read count for cluster inclusion = 1
Minimum read length = 20
Maximum number of non conversion mismatches = 1

## Comparison of eIF3 PAR-CLIP results in Jurkat and HEK293T cells

To compare RNAs in activated Jurkat cells crosslinked to eIF3 with those crosslinked to eIF3 in HEK293T cells (*Lee et al., 2015*), the gene cluster lists (*.gene_cl.csv) from APA_REP1, APA_REP2, APB_REP1, APB_REP2, APD_REP1, and APD_REP2 were used (see *Supplementary file 3*). The genes were first sorted by total read counts ('ReadCountSum') from high to low for each library to obtain the top candidate genes. Then, the same number of top candidate genes from these sorted lists as the number of genes identified in HEK293T cells, for eIF3 subunit EIF3A, EIF3B, and EIF3D crosslinked to RNA (*Lee et al., 2015*), were chosen for comparison. We could not compare RNAs in non-activated Jurkat cells to those in HEK293T cells for the EIF3A/B/C or EIF3D/L samples (NPA_REP1, NPA_REP2, NPD_REP1, NPD_REP2 libraries) due to the low numbers of RNAs that crosslinked to eIF3 in these conditions (157 or 165 RNAs in the EIF3A/B/C samples, and 117 or 126 RNAs in the EIF3D/L samples). In the EIF3B/C samples (NPB_REP1 and NPB_REP2) more RNAs crosslinked to these subunits. However, only 8 RNAs in the NPB_REP1 sample and only 27 RNAs in the NBP_REP2 sample had ReadCountSum values exceeding the threshold used for the APB_REP1 and APB_REP2 libraries from activated cells.

## PAR-CLIP pathway analysis

To determine biological pathways enriched in the set of mRNAs that crosslinked to eIF3 in activated Jurkat cells, genes with at least 100 total aligned reads were used, as determined in the PARpipe analysis described above (*Mukherjee et al., 2019*), from the EIF3A/C/B samples. Since PAR-CLIP

reads are short, it is not possible to determine with certainty which mRNA transcript isoform cross-linked with eIF3. Therefore the most abundant mRNA transcript isoform for each gene was chosen, as determined by transcriptome profiling using kallisto (protein_coding category) (*Bray et al., 2016*), as described in the Transcriptome Profiling section. Even with this choice, eIF3 crosslinks to mRNAs do not correlate with mRNA abundance (*Figure 1—figure supplement 1F*). Human genome GRCh38. p13 annotation was used to extract mRNA lengths by 5'-UTR, coding region and 3'-UTR (Ensembl Biomart) (*Cunningham et al., 2019*). These genes were then sorted by the density of PAR-CLIP reads in the mRNA 5'-UTR region, prior to mapping pathways of transcripts that crosslinked to eIF3. Due to the complexity of *TCRA* and *TCRB* transcript annotation, these transcripts were excluded from the normalization calculation, but included in the pathway analysis. The top 500 genes from the resulting EIF3A/C/B PAR-CLIP gene lists were used, sorted as described above, to analyze gene enrichment profiles in the STRING database (*Szklarczyk et al., 2019*). The top tissue-specific categories involve the immune system and T cell function (*Supplementary file 5*). Note that the STRING database categories are not mutually exclusive gene lists, and do not include TCR subunits in its analysis.

## Metagene analysis

The PAR-CLIP genes sorted as described above in the 'PAR-CLIP pathway analysis' were used and mapped against the most abundant mRNA transcript isoforms to generate cumulative plots of the reads. Reads for the *TCRA* and *TCRB* mRNAs were manually extracted from the Bowtie version 1.0.0 Hg38 alignment of the eIF3 PAR-CLIP reads. We did not identify reads mapped to the D segment of *TCRB* (e.g. to *TRBD2*) due to its short length of 16 nucleotides. These reads were combined with the mapped reads in the *.read.csv files generated by Parpipe. The combined reads were then sorted to extract only reads annotated as 5'-UTR, start codon, coding, stop codon, and 3'-UTR. The most abundant transcript isoform, as identified in the Transcriptome Profiling section using kallisto (described below) was used. Reads mapped to the 5'-UTR and start codon were normalized by the length of the 5'-UTR. Reads mapped to the coding region and stop codon were normalized by the length of the coding region. Finally, reads mapped to the 3'-UTR were normalized to the length of the 3'-UTR. Relative positions of the mapped reads along a metagene were computed based on the locations each mapped read fell within its respective feature. Relative positions were from –1 to 0 for the 5'-UTR, 0 to 1 for the coding region, and 1 to 2 for the 3'-UTR. 5'-UTR values were computed by multiplying the relative position by –1, whereas the 3'-UTR values were computed by adding one to the relative position. Coding region relative positions were unchanged.

The empirical cumulative distribution frequency function from R package ggplot2 (*Wickham and Chang, 2016*) was used to build the metagene plots from the vector of relative positions for the reads which mapped to a given set of reads. We defined mRNAs having a ratio of normalized 5'-UTR reads divided by 3'-UTR reads of 20 or more as '5'-UTR enriched' with respect to eIF3 crosslinking. All others were categorized as 'pan-mRNAs', with eIF3 crosslinking across the entire length of the mRNA. The cut-off value 20 is not a sharp boundary between the two categories of mRNA (*Figure 1—figure supplement 1G*).

## Transcriptome profiling

RNA samples were extracted from non-activated Jurkat cells or Jurkat cells activated for 5 hr with I + PMA, using the Direct-zol RNA Miniprep kit (Zymo Research). The libraries were prepared using TruSeq Stranded RNA LT Ribo-Zero Gold kit (Illumina) following the manufacturer's instructions, with two biological replicates. Cutadapt (version 2.6) (*Martin, 2011*) with a minimum read length of 20, 5' end with a cutoff of 15 and the 3' end with a cutoff of 10 in paired-end mode was used to remove adapters. RNA-seq reads were pseudoaligned using kallisto v.0.46.0 run in quant mode with default parameters to estimate transcript abundance (transcripts per million, TPM) (*Bray et al., 2016*). The transcript index for kallisto was made with default parameters and GENCODE Release 32 (GRCh38. p13) FASTA file (*Frankish et al., 2019*).

## RNA-FISH and immunofluorescence

Jurkat cells were washed with PBS, fixed with 3.7% (vol./vol.) paraformaldehyde (VWR) for 10 min at room temperature and washed three times with PBS. PBS was discarded and 1 ml 70% ethanol was added. The cells were incubated at 4 °C for 16 hr. The 70% ethanol was aspirated and the cells were

washed once with 0.5 ml Stellaris RNA wash buffer A (Biosearch technologies). The cells were then incubated with 100 µl Stellaris hybridization buffer (Biosearch Technologies) containing Stellaris RNA FISH probes (Biosearch Technologies) at a final concentration of 125 nM (*Supplementary file 6*) and with the relevant antibody (*Supplementary file 6*) for 16 hr at 28 °C. The cells were then washed twice with 0.5 ml Stellaris RNA wash buffer A containing secondary antibody conjugated with a fluorophore for 30 min at 37 °C in the dark. The second Stellaris RNA wash buffer A contained DAPI in addition to the secondary antibody. Finally, the cells were washed once with 0.5 mL Stellaris RNA wash buffer B and mounted with mounting solution (Invitrogen). All high-resolution images were taken using confocal ZEISS LSM 880 in Airyscan super-resolution mode, equipped withA Plan-Apochromat 63 x/1.4 Oil objective (Zeiss). To measure colocalization of *TCRA* and *TCRB* mRNAs with each other or with P bodies (using DCP1 antibody, *Supplementary file 6*) or stress granules (G3BP1 antibody, *Supplementary file 6*) the cells were mounted along with 0.1 µm TetraSpeck microspheres (ThermoFisher) adhered to the slide according to manufacturer's instructions, to be able to account for the chromatic shift during image acquisition.

## Colocalization analysis

To measure colocalization of *TCRA* and *TCRB* mRNAs with each other or with P bodies (using DCP1 antibody) or stress granules (G3BP1 antibody), immunofluorescently labeled cells (see above) were mounted along with 0.1 µm TetraSpeck microspheres (ThermoFisher) adhered to the slide according to manufacturer's instructions. The microspheres allowed for the correction of lateral and axial chromatic aberrations during image acquisition.

Z-stacks were acquired with 35 nm x 35 nm x 190 nm voxels on a ZEISS LSM 880 in Airyscan super-resolution mode, equipped with a Plan-Apochromat 63 x/1.4 Oil objective (Zeiss). The images were then deconvolved to a lateral resolution of ≈150 nm and an axial resolution of ≈500 nm (as confirmed by observing the discrete Fourier transform of the z-stacks). After imaging a single cell, beads that were on the slide axial to the cell were imaged to measure the corresponding lateral and axial chromatic aberrations.

To quantify the relative colocalization, we developed an automated processing and analysis pipeline in ImageJ 1.52p available on github: https://github.com/Llamero/TCR_colocalization_analysis-macro (copy archived at swh:1:rev:95714d5f24259dd03aa5c184a15b9d5f1bb17b40, *Marson, 2021*). Specifically, the chromatic aberrations in the z-stacks were compensated for by registering the channels of the bead z-stack to one another, and then applying the same registration vectors to the corresponding channels in the cell z-stack (*Parslow et al., 2014*). Each channel of a z-stack was then thresholded to remove background in the image, and then the colocalization between each pair of images was measured using the Pearson's correlation coefficient. Samples in which any pair of channels in the bead z-stack had a correlation of less than 0.45 were removed from final analysis, as this suggested that the images had insufficient dynamic range in at least one of the channels for an accurate deconvolution.

## Polysome analysis of eIF3-associated mRNAs

To isolate polysomes from Jurkat cells, the cells were seeded to reach $8 \times 10^5$ cells ml$^{-1}$ on the day of harvest and then stimulated with anti-CD3/anti-CD28 antibodies as described above for 5 hr. To isolate polysomes from primary human T cells, the cells were seeded to reach $1 \times 10^6$ cells/mL on the day of harvest and then stimulated with anti-CD3 and anti-CD28 antibodies as described above for 1 hr. Both Jurkat and primary human T cells were treated with 100 µg ml$^{-1}$ cycloheximide (VWR) 5 min before harvesting. Cells were then collected into a 50 ml falcon tube and rinsed once with ice cold PBS (ThermoFisher) supplemented with 100 µg ml$^{-1}$ cycloheximide. The cells were then incubated with 0.5 mM of the crosslinking reagent dithiobis (succinimidyl propionate) (DSP, Thermofisher, Cat. #: PG82081) and 100 µg ml$^{-1}$ cycloheximide in PBS at room temperature for 15 min, with rocking. The crosslinking reagent was then removed and the cells were incubated with quenching reagent (PBS, 100 µg ml$^{-1}$ cycloheximide and 300 mM Glycine(Sigma)) for 5 min on ice. The cells were then rinsed again with ice cold PBS and flash frozen in liquid nitrogen.

A total of $4 \times 10^8$ cells were lysed with 400 µl hypotonic lysis buffer (10 mM Hepes pH 7.9, 1.5 mM MgCl$_2$, 10 mM KCl, 0.5 mM DTT, 1% Triton, 100 µg ml$^{-1}$ cycloheximide, and one Complete EDTA-free Proteinase Inhibitor Cocktail tablet (Roche)). The cells were incubated for 10 min on

ice and then passed through an 18-G needle five times, and centrifuged at 13,000× g for 15 min at 4 °C. The 400 μl supernatant was then transferred to a fresh eppendorf tube and subjected to RNase H digestion by adding the following reagents: 3 mM $MgCl_2$, 10 mM DTT, 200 units RNase H (NEB) and 20 μl of SUPERasIN (ThermoFisher), with a total of 4 μM DNA oligos (IDT), as indicated in the figure legends. The mixture was then incubated at 37 °C for 20 min. After incubation 10 μl of the RNase H treated lysate mixture was isolated to test the efficiency of the RNase H digestion using qRT-PCR (*Figure 2—figure supplement 1B, D and G*) and the rest of the lysate was layered onto a 12 ml 10%–50% sucrose gradient, made with gradient buffer consisting of: 10% sucrose (w/v) or 50% sucrose (w/v), 100 mM KCl, 20 mM Hepes pH 7.6, 5 mM $MgCl_2$, 1 mM DTT and 100 μg ml⁻¹ cycloheximide. The gradient was centrifuged at 36,000 rpm (222 k x g) for 2 hr at 4 °C in a SW-41 rotor. The gradient was then fractionated using the Brandel gradient fractionator and ISCO UA-6 UV detector and all the polysome fractions ( ~ 5 ml) were collected into a fresh 15 ml falcon tube. From each of the polysome fractions, 100 μl was kept aside to measure the input RNA amounts, and the rest of each polysome fraction was incubated with 100 μl of Dynabeads (Invitrogen) conjugated with 40 μl of anti-EIF3B antibody (Bethyl A301-761A) for 16 hr, rotating at 4 °C. After incubation, the beads were rinsed three times with 1000 μl room temperature NP40 lysis buffer (defined in the PAR-CLIP section), rotating for 5 min for each wash. After the final wash the beads were resuspended in 400 μl of Trizol (Thermofisher), the RNA was extracted and qRT-PCR was performed as described above.

## Western blot
Western blot analysis was performed using the antibodies listed in *Supplementary file 6*.

## Total mRNA isolation and quantitative RT-PCR analysis
Total RNA was isolated from whole cells for qRT-PCR using Quick RNA miniprep plus kit from Zymo Research following the manufacturer's instructions. Quantitative RT-PCR analysis was performed using the Power SYBR Green RNA-to-Ct 1-Step kit (Applied Biosystems) according to the manufacturer's instructions, and the QuantStudio 3 Real-Time PCR System (ThermoFisher). Each target mRNA was quantified in three biological replicates, with each biological replicate having three technical replicates.

## Plasmids
Nanoluciferase reporters (*Hall et al., 2012*) were constructed using the 5'-UTR of the human beta globin mRNA (*HBB*) and a PEST destabilization domain. The PEST domain reduces protein half-life (*Voon et al., 2005*) and was used to provide better time resolution of nanoluciferase expression after T cell activation. The *TCRA* 3'-UTR and *TCRB* 3'-UTR sequences were amplified from Jurkat genomic DNA. The nanoluciferease sequence fused to a PEST domain was amplified from pNL1.2[*NlucP*] Vector Sequence (Promega) and was cloned into a modified CD813A vector (System Biosciences) using the In-Fusion HD Cloning Kit (Takara). The subsequent mutations in the *TCRA* and *TCRB* 3'-UTRs were generated using these initial constructs. For *TCRA ΔPAR* constructs, nucleotides 102–338 in the 3'-UTR of *TCRA* mRNA were deleted. For *TCRB ΔPAR* constructs, nucleotides 16–161 in the 3'-UTR of *TCRB* mRNA were deleted. *TCRA/TCRB ΔPAR*, *TCRA/TCRB R*PAR, 3'-LTR* (3'-Long Terminal Repeat), *JUN* 5'-UTR hairpin (*Supplementary file 6*) and HCV IRES domain III (*Supplementary file 6*) sequences were purchased as gblocks from IDT and were cloned into this plasmid backbone. The *WPRE* (Woodchuck Hepatitis Virus Posttranscriptional Regulatory Element) 3'-UTR sequence was amplified from the CD813A-1 (System Biosciences) vector.

For nanoluciferase reporters designed to be membrane-tethered, we fused the N-terminal sequence of CD3-zeta spanning the transmembrane helix (amino acids 1–60) ordered as a gblocks from IDT to the nanoluciferase sequence above. To prevent interaction of the CD3-zeta-nanoluciferase fusion protein with the TCR, we made mutations in the CD3-zeta-derived transmembrane helix that would disrupt interactions with the TCR, based on the cryo-EM structure of the complex (*Dong et al., 2019*) (PDB entry 6JXR) and consistent with earlier biochemical results (*Call et al., 2002*). The two mutations, L35F and D36V, are predicted to introduce a steric clash and disrupt an intra-membrane salt bridge, respectively, with other subunits in the TCR holo-complex. These CD3-zeta-nanoluciferase chimeras were cloned into the modified CD813A plasmids described above.

## Generation of primary human T cells stably expressing nanoluciferase reporters

For lentiviral production, HEK293T cells were plated at a density of 80% in T-75 flasks the night before transfection. The cells were then transfected with plasmids: expressing the nanoluciferase, PsPAX2 and pCMV-VSV-G using the Lipofectamine 2000 reagent (ThermoFisher) following the manufacturer's instructions. Forty-eight hours after transfection, the viral supernatant was collected, filtered, and concentrated using PEG-it Virus Precipitation Solution (System Biosciences) following the manufacturer's instructions. The virus pellets were then resuspended in ImmunoCult-XF T Cell Expansion media and stored in –80 °C.

The primary human T cell transductions were done with multiple viral titers using TransDux MAX Lentivirus Transduction Reagent (System Biosciences) following the manufacturer's instructions. To test the viral transduction efficiency of the cells, 48 hr after viral transduction the percent of cells expressing GFP was measured by FACS analysis and cells expressing less than 30% GFP were treated with 1 µg ml$^{-1}$ puromycin (ThermoFisher) for 4 days or until ~90% of the cells were GFP positive.

## Luciferase reporter assays

The stable cell lines expressing the nanoluciferase reporters were stimulated with anti-CD3/anti-CD28 antibodies with 50 U/mL IL-2 and the nanoluciferase activity was assayed after 30 min, 1 hr, 3 hr, and 5 hr after stimulation using Nano-Glo Luciferase Assay System (Promega). For each time point, 200,000 cells were tested in triplicate for each cell line.

For assays of TCR and CD28 signaling requirements, the stable cell lines were stimulated with anti-CD3 or anti-CD28 antibodies individually in the presence of 50 U/mL IL-2, and assayed as described above. To identify signaling pathways downstream of CD28, cells were incubated with AKT inhibitor AZD5363 at 1 µM (Cayman Chemical), mTOR inhibitor Torin 1 at 250 nM (Cayman Chemical) or DMSO for 3 hr prior to T cell activation with anti-CD3/anti-CD28 antibodies (Tonbo) as indicated above. The cells were assayed as described above, and the extent of AKT and mTOR inhibition were determined by western blot analysis of phosphorylation of their substrates GSK-3β and 4E-BP1, respectively.

## RNA immunoprecipitation and qPCR

The EIF3B-RNA immunoprecipitations were carried out following the exact same conditions used for the PAR-CLIP analysis with the following changes. For each immunoprecipitation, cell lysates were prepared in NP40 lysis buffer (defined in the PAR-CLIP section) with 4 million cells. The lysates were then incubated with 50 µl Protein G Dynabeads conjugated with 20 µl of anti-EIF3B antibody (Bethyl A301-761A) for two hours at 4 °C. After incubation, the flow through was removed and the beads were washed three times with 1 ml NP40 lysis buffer for each wash. The beads were then resuspended in 400 µl TRIzol reagent (ThermoFisher) and vortexed for 1 min. The RNA was extracted following the manufacturer's instructions and qPCR was performed as described above using primers listed in *Supplementary file 6*.

## sgRNA/Cas9 RNP production

The sgRNA/Cas9 RNPs used to edit Jurkat cells were produced by complexing sgRNA (Synthego) to Cas9 as described (*Schumann et al., 2015*) while RNPs to edit Primary Human T cells were produced by complexing a two-component gRNA (crRNA and tracrRNA, Dharmacon) to Cas9 as described in *Roth et al., 2018*. The targeting sequences for the sgRNAs and crRNA are given in *Supplementary file 6*. Recombinant Cas9-NLS was obtained from MacroLab in the California Institute for Quantitative Biosciences.

## Primary T cell and Jurkat genome editing

Jurkat cells used for electroporation were collected at passage five or lower and were maintained at a seeding density of 8 × 10$^5$ cells ml$^{-1}$ or lower. Primary T cells were isolated as described above. Prior to electroporation the Primary T cells were stimulated with magnetic anti-CD3/anti-CD28 Dynabeads (ThermoFisher) for 48 hr. After 48 hr, these beads were removed from the cells by placing cells on an EasySep cell separation magnet for 2 min before electroporation. One million Jurkat (not activated) or primary T cells cells (activated with anti-CD3/anti-CD28 Dynabeads for 48 hr) were rinsed with PBS and then resuspended in 20 µl of Lonza electroporation buffer P3. The cells were then mixed with

2.5 µl Cas9 RNPs (50 pmol total) along with 2 µl of a 127-nucleotide non-specific single-stranded DNA oligonucleotide at 2 µg µl⁻¹ (4 µg ssDNA oligonucleotide total). The cells were then electroporated per well using a Lonza 4D 96-well electroporation system with pulse code DN100 for Jurkat cells and EH115 for primary human T cells. Immediately after electroporation, 80 µl of pre-warmed media (without cytokines) was added to each well, and the cells were allowed to rest for 15 min at 37 °C in a cell culture incubator while remaining in electroporation cuvettes. After 15 min, cells were moved to final culture flasks. Jurkat cells were clonally selected by single cell sorting into U-bottomed 96-well plates and by testing each clone using PCR primers flanking the editing site (*Figure 5—figure supplement 1A*). The clones producing a single PCR band of 1283 bp and 1022 bp were selected as clonal populations for *TCRA ΔPAR* and *TCRB ΔPAR,* respectively.

Genome edited populations of primary T cells with the *TCRA ΔPAR* and *TCRB ΔPAR* mutations were determined by measuring the density of the PCR bands described above resulting from the edited cell population compared to the PCR band from non edited cells, using ImageJ. To compare with the *TCRA ΔPAR* or *TCRB ΔPAR* primary T cell populations, we edited cells from both donors using each gRNA targeting the *TCRA* 3'-UTR and *TCRB* 3'-UTR region separately (single gRNA experiments), a gRNA targeting the coding sequence (CDS) of *TCRA* which knocks out TCR expression with high efficiency (Δ*TCR*) (*Roth et al., 2018*), a scrambled gRNA (SC) which does not target any site in the human genome, and cells mixed with the gRNA/Cas9 RNPs but not nucleofected.

## Analysis of TCR cluster formation

WT, *TCRA ΔPAR,* or *TCRB ΔPAR* T cells were activated with anti-CD3/anti-CD28 antibodies for 1, 3, or 5 hr, as described above. Cells were collected and stained with anti-TCRA antibodies, followed by a secondary antibody Alexa Fluor 488 goat anti-mouse IgG (Invitrogen). For the counting of cells containing TCR clusters, immunofluorescent imaging was performed on a Revolve Epi-Fluorescence microscope (Echo), equipped with an A Plan-Apochromat 40 x objective (Olympus). Cells with substantial puncta (arrows in *Figure 5—figure supplement 1C*) were scored as having TCR cluster formation (*Figure 5C*).

## Flow cytometry and cell sorting

Flow cytometric analysis was performed on an Attune NxT Acoustic Focusing Cytometer (ThermoFisher). Surface staining for flow cytometry and cell sorting was performed by pelleting cells and resuspending in 50 µl of FACS buffer (2% FBS in PBS) with antibodies at a 1:100 concentration (*Supplementary file 4*) for 20 min at 4 °C in the dark. Cells were washed twice in FACS buffer before resuspension and analysis. For the analysis of cell surface TCR levels, the gate for TCR+ cells was set based on analysis of TCR levels in the TCR knockout (Δ*TCR*) T cells. The gates for CD69+ and CD25+ cells were set based on CD69 and CD25 levels in the Δ*TCR* T cells.

## ELISA

The cell culture supernatants were collected after each time point of activation with anti-CD3/anti-CD28 antibodies for WT, *TCRA ΔPAR,* or *TCRB ΔPAR* cells. For each timepoint, the same number of cells were used from each strain to be able to compare across strains and time points. The amount of secreted IL-2 in the cell suspensions after activation anti-CD3/anti-CD28 antibodies for WT, *TCRA ΔPAR* or *TCRB ΔPAR* cells were measured by ELISA MAX Deluxe Set Human IL-2 (BioLegend) according to the manufacturer's instructions.

## Chimeric antigen receptor (CAR) construct sequences

The CDS region of the Juno anti-CD19 chimeric antigen receptor (*June et al., 2014*; *Kalos et al., 2011*) or the Kite anti-CD19 chimeric antigen receptor (*Kochenderfer et al., 2010*; *Kochenderfer et al., 2009*) (GenBank entry HM852952.1), both presently used in the clinic, were cloned into CD813A lentiviral vectors (Systems Biosciences) with a common core EF1alpha promoter and 5'-UTR with an inserted intron. The CDS sequence was followed by various 3'-UTRs. Two of the 3'-UTRs are presently used in clinical CAR T cells, the Woodchuck Hepatitis Virus Posttranscriptional Regulatory Element (*WPRE*) (*June et al., 2014*; *Milone et al., 2009*), or the murine stem cell virus (MSCV) retroviral 3'-long terminal repeat (*3'-LTR*) (*Kochenderfer et al., 2009*). We also cloned the full *TCRA* or *TCRB* 3'-UTR sequences including the polyadenylation sites after the anti-CD19 CAR CDS sequence.

## Production of CAR T cells

Production of the CAR-expressing lentiviruses was carried out as described above using HEK293T cells. The viruses were then concentrated using PEG-it Virus Precipitation Solution (System Biosciences) following the manufacturer's instructions. The virus pellets were then resuspended in ImmunoCult-XF T Cell Expansion media and stored in –80 °C. Frozen primary human T cell pellets were thawed and then stimulated at a starting density of approximately 1 million cells per mL of media with ImmunoCult Human CD3/CD28/CD2 T Cell Activator (Stemcell) for 48 hr and then transduced with various CAR viruses using TransDux MAX Lentivirus Transduction Reagent (System Biosciences) as described above. Two days after transduction the percent of cells expressing GFP was measured by FACS analysis and cells expressing less than 30% GFP were treated with 1 µg mL$^{-1}$ puromycin (ThermoFisher) for 2 days or until ~90% of the cells are GFP positive. After removal of puromycin the cells were seeded at $1 \times 10^6$ cells/mL up to 9 days from the day the cells were transduced by adding fresh ImmunoCult-XF T Cell Expansion Medium (StemCell) and fresh IL-2 (500 U/mL or 50 U/mL) every other day. The amount of IL-2 added varied based on the experiment performed as mentioned in the figure legends. After 9 days, the CAR T cells were used for various assays.

## Production of TCR knockout CAR T cells

Frozen primary human T cell pellets were thawed and then stimulated at a starting density of approximately 1 million cells per mL of media with anti-human CD3/CD28 Dynabeads (ThermoFisher), at a bead to cell ratio of 1:1, and cultured in ImmunoCult-XF T Cell Expansion media containing IL-2 (500 U ml$^{-1}$;Peprotech), IL-7 (5 ng ml$^{-1}$; ThermoFisher), and IL-15 (5 ng ml$^{-1}$; Life Tech). After 24 hr of stimulation, cells still attached to the Dynabeads were transduced with various CAR lentiviruses using TransDux MAX Lentivirus Transduction Reagent (System Biosciences) as described above. After 48 hr of stimulation (i.e. 24 hr after viral transduction), the beads were removed from the cells by placing cells on an EasySep cell separation magnet for 2 min before electroporation. One million CAR T cells were then electroporated with Cas9 RNP targeting the *TCRA* CDS (**Supplementary file 6**) using the pulse code EH115 as described above. Immediately after electroporation, 80 µL of pre-warmed media was added to each well and incubated at 37 °C for 20 min. Cells were then transferred to a round bottom 96-well plate with 50 U/mL IL-2 at $1 \times 10^6$ cells /mL in appropriate tissue culture vessels. Three days after the nucleofection, the percent of cells expressing GFP was measured by FACS analysis and cells expressing less than 30% GFP were treated with 1 µg mL$^{-1}$ puromycin (ThermoFisher) for 2 days or until ~90% of the cells were GFP positive. After removal of puromycin the cells were seeded at $1 \times 10^6$ cells/mL up to 9 days from the day the cells were transduced by adding fresh ImmunoCult-XF T Cell Expansion Medium (StemCell) and fresh IL-2 (50 U/mL) every other day. After 9 days the CAR T cells were used for various assays.

## CAR expression dynamics

To measure CAR expression dynamics by western blot analysis, CAR T cells were incubated with NALM6 tumor cells at a ratio of 1:2 CAR T cell:NALM6 cells, in ImmunoCult-XF T Cell Expansion Medium (StemCell) and 50 U/mL IL-2. Cells were collected at the indicated time points and processed for western blot analysis as described above. The total protein expression level of the anti-CD19 CAR was detected with an anti-CD3z antibody (Santa Cruz Biotechnology).

## CAR T-cell cytotoxicity assays

CAR T-cell cytotoxicity was determined using a FACS based assay. First, NALM6 or Jeko 1 tumor cells were stained with CTV (CellTrace Violet, Thermofisher). Cells were resuspended in 10 mL PBS ($5 \times 10^6$ cells), 5 µl CTV was added, and the cells were incubated for 20 min at 37 °C. After incubation, 30 mL of RPMI media was added and the cells incubated for an additional 5 min at 37 °C. The cells were then gently pelleted, rinsed and resuspended in fresh ImmunoCult-XF T Cell Expansion Medium (StemCell) and 50 U/mL IL-2. Then in a round bottom 96-well plate, 20,000 stained tumor cells were cocultured with CAR T cells at different effector-target ratios in a total volume of 200 µL for 24, 48, and 72 hr. T cells transduced with a membrane-tethered nanoluciferase reporter containing the *TCRA* 3'-UTR (**Figure 4B**) were used as a negative control. The CAR T cell killing capacity was measured by Flow Cytometry as diagrammed in **Figure 6—figure supplement 1B**. Briefly, the 96-well plates were incubated at 4 °C for 30 min to terminate the killing by CAR T cells. Then 2 µL of propidium iodide

(1000 x stock, Thermofisher) was added to each column in the plate one by one and the dead and living tumor cells, as well as live T cells, were measured as shown in *Figure 6—figure supplement 1B*.

## Acknowledgements

We thank M Hafner for advice on PAR-CLIP methodology and data analysis, A Weiss for experimental suggestions and advice, H Nolla and A Valeros at the UC Berkeley flow cytometry facility for helping out with the FACS analysis and single cell sorting, F Ives and H L Aaron at the UC Berkeley Imaging center for help with microscopy, J Lui and J Bohlen for advice on planning experiments and for suggestions on the manuscript, N Aleksashin and AM González-Sánchez for advice on the manuscript, and M Mignardi for advice on FISH experiments. *Figures 6E and 7* were created using Biorender.

## Additional information

### Competing interests

Dasmanthie De Silva, Jamie HD Cate: A provisional patent application has been filed on some of the work presented herein. Theodore L Roth: is a co-founder of Arsenal Therapeutics. Alexander Marson: is a compensated co-founder, member of the boards of directors, and a member of the scientific advisory boards of Spotlight Therapeutics and Arsenal Biosciences. Was a compensated member of the scientific advisory board at PACT Pharma and was a compensated advisor to Juno Therapeutics. Owns stock in Arsenal Biosciences, Spotlight Therapeutics, PACT Pharma and Merck. AM has received fees from Vertex, Merck, Amgen, Trizell, Genentech, AlphaSights, Rupert Case Management and Bernstein. Is an investor in and informal advisor to Offline Ventures. The Marson lab has received research support from Juno Therapeutics, Epinomics, Sanofi, GlaxoSmithKline, Gilead, and Anthem. The other authors declare that no competing interests exist.

### Funding

| Funder | Grant reference number | Author |
|---|---|---|
| National Institutes of Health | R01-GM065050 | Dasmanthie De Silva<br>Grant H Chin<br>Jamie HD Cate |
| Tang Prize for Biopharmaceutical Science | | Jamie HD Cate |
| Damon Runyon Cancer Research Foundation | DRR#37-15 | Nicholas T Ingolia |
| National Institutes of Health | DP2 CA195768 | Lucas Ferguson<br>Marek Kudla<br>Nicholas T Ingolia |
| National Institutes of Health | P30EY003176 | Benjamin E Smith |
| National Institutes of Health | S10 OD018174 | Dasmanthie De Silva |
| Care-for-Rare Foundation | | Franziska Blaeschke |
| German Research Foundation | | Franziska Blaeschke |
| Burroughs Wellcome Fund | | Alexander Marson |
| Cancer Research Institute | | Alexander Marson |
| Chan Zuckerberg Initiative | | Ryan A Apathy<br>Theodore L Roth<br>Alexander Marson |
| Innovative Genomics Institute | | Ryan A Apathy<br>Theodore L Roth<br>Alexander Marson |

| Funder | Grant reference number | Author |
|---|---|---|
| Parker Institute for Cancer Immunotherapy | | Alexander Marson |

The funders had no role in study design, data collection and interpretation, or the decision to submit the work for publication.

## Author contributions

Dasmanthie De Silva, Conceptualization, Data curation, Formal analysis, Investigation, Methodology, Project administration, Supervision, Validation, Visualization, Writing – original draft, Writing – review and editing; Lucas Ferguson, Grant H Chin, Formal analysis, Investigation, Methodology, Visualization, Writing – review and editing; Benjamin E Smith, Formal analysis, Methodology, Software, Supervision, Visualization, Writing – review and editing; Ryan A Apathy, Franziska Blaeschke, Investigation, Methodology, Resources, Writing – review and editing; Theodore L Roth, Investigation, Methodology, Resources; Marek Kudla, Data curation, Formal analysis, Methodology, Writing – review and editing; Alexander Marson, Funding acquisition, Methodology, Resources, Supervision, Writing – review and editing; Nicholas T Ingolia, Formal analysis, Funding acquisition, Investigation, Methodology, Supervision, Writing – review and editing; Jamie HD Cate, Conceptualization, Formal analysis, Funding acquisition, Investigation, Methodology, Project administration, Supervision, Writing – original draft, Writing – review and editing

## Author ORCIDs

Dasmanthie De Silva http://orcid.org/0000-0001-6824-0850
Nicholas T Ingolia http://orcid.org/0000-0002-3395-1545
Jamie HD Cate http://orcid.org/0000-0001-5965-7902

## Decision letter and Author response

Decision letter https://doi.org/10.7554/eLife.74272.sa1
Author response https://doi.org/10.7554/eLife.74272.sa2

---

# Additional files

## Supplementary files

• Supplementary file 1. Subunits in eIF3 crosslinked to RNA in activated Jurkat cells. Lists the eIF3 subunits, percent sequence coverage and number of identified peptides.

• Supplementary file 2. PARpipe statistics for eIF3 PAR-CLIP samples. Samples are indexed in the first tab, including both biological replicates for activated and non-activated Jurkat cells. Statistics are given for each library at the read, cluster and group level.

• Supplementary file 3. PAR-CLIP mapping to individual genes for each eIF3 PAR-CLIP sample. Samples are indexed in the first tab, including both biological replicates for activated and non-activated Jurkat cells. First lists the gene name and number of clusters identified. The statistics also include: Sum, sum of that statistic over all sites for that gene; Med, median of that statistic for all sites for that gene; ReadCount, total reads mapping to the gene; T2C fraction, number of reads with T-to-C conversions / number of reads; ConversionSpecificity, log (number of reads with T-to-C conversions / number of reads with other conversions); UniqueReads, reads collapsed to single copies. Also included: 5'utr/Intron/Exon/3'utr/Start_codon/Stop_codon, number of sites mapping to that annotation category; Junction, number of sites mapping to a junction between categories (coding-intron, coding-3'utr, etc.); GeneType, as described in the gene_type category for this gene in the.gtf file used.

• Supplementary file 4. Transcriptome analysis of non-activated or activated Jurkat cells. Each tab lists transcript name and version, gene name, type of transcript, length of transcript, and mean transcripts per million, calculated from two biological replicates.

• Supplementary file 5. Pathway enrichment analysis. Lists for both biological replicates of the EIF3A/C/B PAR-CLIP libraries are included (genes with ≥ 100 reads), along with associated transcript names, lengths in nts of the 5'-UTR, coding region, and 3'-UTR, and reads normalized to the lengths of the transcript regions. Tabs also include the top tissue-specific pathway enrichment categories determined using the STRING Database. These list: the Gene Ontology (GO) number, GO description, observed gene count, background gene count, false discovery rate, and matching

proteins in the network by Ensembl protein ID, and by gene name.

• Supplementary file 6. Reagent information for experiments. Lists include antibodies used, PCR primers, qPCR primers, gRNA targeting sequences, and FISH probes, and DNA oligos for RNase H experiments.

• Transparent reporting form

• Source data 1. Original western blots for main figures and figure supplements. Boxed regions are the regions shown in the figures.

• Source data 2. Spreadsheets with numerical data used in figures and figure supplements.

## Data availability

Sequencing data has been deposited in GEO (GSE191306). Code used to analyze the microscopy images is available on github at https://github.com/Llamero/TCR_colocalization_analysis-macro (copy archived at https://archive.softwareheritage.org/swh:1:rev:95714d5f24259dd03aa5c184a15b9d5f1bb17b40).

The following dataset was generated:

| Author(s) | Year | Dataset title | Dataset URL | Database and Identifier |
|---|---|---|---|---|
| De Silva D, Cate JH, Chin G | 2021 | Genome-wide mapping of eIF3-RNA interactions in Jurkat cells using PAR-CLIP | https://www.ncbi.nlm.nih.gov/geo/query/acc.cgi?acc=GSE191306 | NCBI Gene Expression Omnibus, GSE191306 |

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
