## [Editor Report]

The work addresses the role of translation initiation in the early induction of specific genes during T cell activation. It primarily uses the Jurkat T cell line and demonstrates that the translation initiation factor eIF3 interacts with the 3'-untranslated regions of the TCRA and TCRB mRNAs. This interaction resulted in a rapid burst in TCRA and TCRB translation through a mechanism dependent on CD28. Adding the TCRA or TCRB 3' UTR to an anti-CD19 chimeric antigen receptor (CAR) improves the ability of the corresponding CAR-T cells to kill tumor cells in vitro.

---

## [Decision Letter]

**Decision letter after peer review:**

Thank you for submitting your work entitled "Robust T cell activation requires an eIF3-driven burst in T cell receptor translation" for consideration by *eLife*. Your article has been reviewed by 3 peer reviewers, and the evaluation has been overseen by Dr B. Malissen as the Reviewing Editor and Tadatsugu Taniguchi as the Senior Editor. The reviewers have opted to remain anonymous.

The reviewers have discussed the reviews with one another and the Reviewing Editor and their views concur.

The work primarily uses the Jurkat T cell line and demonstrates that eIF3 binds to TCRA and TCRB RNAs. The 3' UTR of TCRA and TCRB was responsible for eIF3 binding and this interaction resulted in a rapid burst in TCRA and TCRB translation through a mechanism dependent on CD28. Adding the TCRA or TCRB 3' UTR to a CAR-T construct improved its ability to kill tumor cells. Although the work has potential for publication in *eLife*, it requires essential additional data to support the central claims of the paper. Each reviewer raised substantive concerns (see below) that need to be resolved experimentally. To quote a few, you should document in Jurkat and in primary T cells whether the rapid burst in TCRA and TCRB translation results in TCR up regulation on the cell surface. This would be important for the functional implications of your findings and considering that (1) CD247 is generally considered as the limiting factor in TCR surface expression, and (2) primary T-cells display a phenomenal sensitivity towards antigen, as the presence of typically only a few antigens on the APC is sufficient to trigger the full T-cell response. Along the same line the readout of T-cell activation shown in Figure 5 and 6 are rather weak. For instance, in Figure 6, gating for CD25 and CD69 is hard to see in the raw flow cytometry plots. Can this be adjusted with the biexponential axis? How were these gates set? In Figure 5, providing primary where possible will support results represented purely as graphs. Among the mechanistic issues, one particularly puzzling issue concerns the observation that the CD28-dependent burst in translation requires membrane tethering whereas the CD3-dependent increase does not. Please note that during the Reviewer discussion, it has been noted that the Western blots displayed in Figure 6D showed strikingly similar bands for donors 1 and 2 in several of the rows probed for CAR-TCR or HSP90. Even the slopes and shapes of these band seem similar. Accordingly, Reviewers have suggested to include the original blots in the revision.

*Reviewer #1:*

Da Silva et al. demonstrate via biochemical and functional assays and bioinformatics that the 3'UTR of both TCRalpha and β are targeted as part of translationally active polysomes by the eukaryotic initiation factor 3 (elf3) in a fashion that depends on the activation of the costimulatory CD28 signaling pathway. They also show by means of mutagenesis of the 3'UTRs that translation of the resulting TCR mRNAs is increased under conditions where elf3 is allowed to bind. By means of nanoluciferase-based reporter assays they also provide evidence that 3'UTR-targeting by elf3 depends on membrane-proximal CD28 signaling. Furthermore, they were able to link activation-driven burst of TCR-translation levels to elf3-3'UTR binding. They present a convincing case that elf3-3'UTR binding in response to TCR-mediated stimulation promotes TCR-clustering, upregulation of CD25 and CD69 (in CD8^+^ T-cells) as well as IL-2 (and to some extent IFNγ) secretion. Lastly, they show that CD19-CARs equipped with either a TCRa- or a TCRb-3'UTR are earlier and more robustly translated upon CAR-T-cell activation with a B-cell lines. This behavior coincides with a somewhat increased CAR-T-cell-mediated killing activity especially at lower CAR-T-cell:target cell ratios or when CAR-T-cells are confronted with a target cell line expressing lower levels of CD19. Overall the study highlights the importance of the 3'UTR for TCR and CAR translation in a setting of CD28-signaling with implications for (CAR-)T-cell functionality.

Strengths

Apart from the overall subject being highly relevant for a better understanding of T-cell physiology, T-cell antigen recognition and the design and implementation of improved CAR-T-cell-based immunotherapies, the study excels with the quality of experimentation and the quantitative nature, its readability and clear conclusions that are in my view nuanced enough to avoid overinterpretation of the findings. To my best knowledge methodologies and the execution of the experiments are all described with necessary clarity and precision leaving little to no room for ambiguities, so that others in the field will be able to reproduce the data.

I find both the reasoning underlying the flow of experimentation and the conclusions drawn from individual observations convincing and compelling. Leads from bioinformatics are verified by detailed biochemistry with clean and clear-cut results, which are then followed up with (i) functional studies concerning mRNA-stability, -elf3 binding and -translation and (ii) studies pertaining to T-cell physiology, all of which provide insights of high relevance.

The highly quantitative nature of all readouts applied (which all include meaningful negative controls) leaves little to no doubts with regard to the conclusions drawn.

Weaknesses

I find only few weaknesses which pertain to the (i) proposed mechanism by which CD28-downstream signaling affects elf3-binding to the 3'UTRs and (ii) the readout of TCR-surface expression and also T-cell activation, the latter lacking definition with regards to TCR-/CAR-proximity. Keeping the scope of the study on in mind, I am of the opinion that these subjects warrant in large part a more detailed discussion and in some cases a few more experiments which could be done with reasonable effort.

(i) Worthy of discussion: To me it is not clear how the CD28/PI3K pathway affects elf3-3'UTR binding. To what extent the binding elf3 to other parts of the mRNA effected by this pathway? Is this a general effect of one that is specific to selected 3'UTRs. How could this be addressed in future experiments? It has been reported that the PD1-PD-L1 interaction as it often occurs in the tumor microenvironment or in another setting of T-cell exhaustion affects predominantly CD28 signaling. Is the PD1-mediated inhibition of the T-cell response directly linked to a loss in elf3-3'UTR binding?

(ii) The authors show a boost in TCR and CAR translation that depends on the 3'UTR being able to bind elf3. How is TCR- and CAR- surface expression affected in the course of activation? How is activation-induced TCR- and CAR- downregulation affected? Primary T-cells feature a phenomenal sensitivity towards antigen, as the presence of typically only a few antigens on the APC is sufficient to trigger the full T-cell response. The situation with CARs is different in that they require 100x – 1000x more antigen for stimulation. How is (CAR-)T-cell sensitivity affected by elf3-3'UTR binding? Time-lapsed flow cytometry experiments using IP26/recombinant CD19 (both commercially available) for detection and a simple titration experiment (involving plate-bound OKT3 or CD19 for stimulation) would clarify these sets of questions.

I am fond of this manuscript. because it provides an important aspect of TCR- and CAR-related physiology that appears highly relevant for T-cell-related immune protection and pathologies as well as T-cell based immunotherapies. On the one hand, it becomes clear now, that attention will need to be paid when exchanging TCRs via CRISPR-Cas9-based approaches. On the other hand, CARs featuring a TCRa or TCRb 3'UTR may give rise to better overall performance (to be tested in more detail in vitro and in vivo).

As I have outlined in previous sections of my review I regard the quality of the presented work as high. I also agree with the flow of experimentation and the conclusions drawn. What is missing in my view is a more detailed investigation (within the scope of this manuscript.) or discussion of the underlying mechanisms (see suggestions in the previous section). I recommend accepting the manuscript. with minor revisions.

Suggested points:

(i) Discuss and assess in more detail how CD28-downstream signaling (PI3K activity) may affect elf3-3'UTR binding.

Are Tec-kinases, which are known to boost TCR-downstream signaling, involved? An experiment involving pharmacological Tec-inhibition would be insightful (applying 3'UTR mutants).

(ii) TCR-/CAR-surface expression: how is it affected by the nature of the TCRa,b 3'UTR before, during and after activation? A simple flow cytometry experiment would tell. For T-cells OKT3 could be used for stimulation and IP26 for surface detection. For CD19-CAR-T-cells, B-cell lines could be used for stimulation and recombinant CD19 for CAR-detection.

(iii) One last aspect that would boost in my opinion the significance of the findings is antigen sensitivity, which could be addressed with little (to reasonable) effort by subjecting (CAR-) T-cells to antigen (OKT3, CD19) immobilized in a titrated fashion onto plates. Readout of activation could be as simple as measuring cytokine secretion in the tissue culture supernatant.

*Reviewer #2:*

The authors explore the role of eIF3 in T cells using a wide variety of experimental techniques. Experiments are very well-explained in the text and documented in the methods. They begin by using PAR-CLIP in the Jurkat T cell line and find that eIF3 binds to TCRA and TCRB, among other RNAs. They also identify a new pattern of eIF3 binding to the coding sequences of many genes, some of which are validated in primary human cells. Looking at the 3' UTR of TCRA and TCRB, the authors show that eIF3 binding to the 3' UTRs is responsible for activation-induced translational bursting, through a mechanism dependent on CD28. Finally, the authors add the TCRA or TCRB 3' UTR to a CAR-T construct and test its impact on CAR expression and cancer cell killing. Some findings in the study are better-supported than others. The finding of eIF3 controlling translational bursting of TCRA and TCRB by binding the 3' UTR is convincing. However, the discovery of pan-mRNA binding behaviour of eIF3 could use more support from primary cells, and the data from CAR-T cell experiments is subtle and low in replicates. For these reasons it is hard to interpret the data.

1) The use of a cell line in PAR-CLIP discovery experiments detracts from the generalizability of results, particularly the new finding of coding sequence binding. While the validation of several transcripts in primary human cells is appreciated, it is hard to know how common this phenomenon is. It is also unfortunate that TCRA and TCRB could not be validated in primary cells.

2) When looking at transcript localization in Jurkat cells by FISH, all the mRNA transcripts measured were in distinct puncta. Is this common for all transcripts? It would be important to see a transcript that behaves differently in order to conclude that the puncta are significant.

3) CAR T cell data is one representative of 2 donors, and the killing enhancement seen with TCRA or TCRB 3' UTR addition is mild. More biological replicates would be needed to convincingly show this with statistical testing.

Recommendations for the authors

1) It is possible to expand primary cells to 55 million, and this would enable the PAR-CLIP to be carried out on primary cells.

2) The authors conclude that eIF3 binding is specific to selected transcripts based on the fact that eIF3 cross-linking isn't correlated with mRNA abundance. Are these selected transcripts simply those that are newly transcribed? This would also explain why the eIF3-bound transcripts are highly enriched for immune processes.

3) The comparison of PAR-CLIP results between HEK293T and Jurkat cells relies on ranking and thresholding. It would be preferable to do a more formal statistical comparison.

i) In several places in the Methods there seem to be conflicting descriptions of Jurkat cells being used at 8 x 105 versus 8 x 106 / mL.

4) Gating for CD25 and CD69 is hard to see in the raw flow cytometry plots. Can this be adjusted with the biexponential axis? How were these gates set? Many percentages on gates in Figure 5 —figure supplement 2a are missing.

5) The live cell gate in Figure 6 —figure supplement 1b is non-conventional, at least to this reviewer. Can this be explained?

*Reviewer #3:*

De Silva et al. 2021 present exciting results regarding the role of the eIF3 complex in T cell activation. The authors use PAR-CLIP to identify the binding patterns of eIF3 in Jurkat cells and find that regions within the 3´ UTRs of TCRA and TCRB mRNAs are sufficient to increase their translation and drive T cell activation. Moreover, expression of the TCRA and TCRB 3´UTRs in CAR T cells improves their ability to kill tumor cells, demonstrating the therapeutic potential of their findings.

– Figure 1: Venn diagrams should include data from non-activated Jurkat cells. The overlap between different data sets (eg. EIF3A/C/B vs. EIF3D/L) should also be represented using a Venn diagram and included in this figure to convince the reader that the IPs against the different subunits are representative of what the entire complex is binding to. GO terms for overlapping mRNAs should be moved to the main figure and discussed in more detail in the text.

– All sequencing tracks presented in the manuscript should include a scale in order for the reader to better assess the data.

– When discussing the PAR-CLIP data, the authors state that "eIF3 crosslinking does not correlate with mRNA abundance." Was RNA-seq performed in parallel? These data should be discussed in more detail and the relevant analyses should be included in the main figures.

– The authors specify that the sites that were modified within the 3´UTRs of TCRA and TCRB were identified through PAR-CLIP. However, the size of the deleted sequence is only mentioned in the context of their CRISPR/Cas9 deletion mutants (Figure 5 supplement 1A). Was the same region targeted in their reporters? Is there evidence that TCRA and TCRB mRNAs that contain these deletions in their 3´UTRs are adequately exported out of the nucleus?

– The data presented in figure 2 nicely demonstrate that eIF3 is bound to the coding regions of certain mRNAs during their translation. However, it does not provide evidence that the binding of eIF3 occurs "via" the coding sequence as the authors claim in the Discussion section of the manuscript.

– The lack of colocalization with stress granules and P bodies alone does not provide convincing evidence that the TCRA and TCRB are in "translational hot spots." Figure 1F should be moved to the supplements as supporting information for the sucrose gradients presented in figure 2.

– In figure 3E, the authors demonstrate that the 3´UTRs of TCRA and TCRB are required for the interaction between these RNAs and eIF3. What is the proposed mechanism for interactions between eIF3 and the 3´UTR that appear to drive the extensive interaction of eIF3 with the mRNA and the observed translational increase?

– How does eIF3 spread across the CDS? What is the relevance of the interactions between eIF3 and the CDS?

– The authors should discuss why the CD28-dependent burst in translation requires membrane tethering but the CD3-dependent increase does not. This section emerged from nowhere and greater explanation would be helpful both for the technical details and for the potential physiological importance of the membrane tethering dependency.

– De Silva et al. demonstrate that expressing the 3´UTRs of TCRA and TCRB in CAR T cells leads to a faster increase in CAR protein expression. However, their data indicate that the increase in translation is transient (Figure 6D and Figure 6 supplement 1A). Therefore, the authors should comment on whether this decreased lag time in protein expression might also accelerate CAR T cell exhaustion.

[Editors' note: further revisions were suggested prior to acceptance, as described below.]

Thank you for resubmitting your work entitled "Robust T cell activation requires an eIF3-driven burst in T cell receptor translation" for further consideration by *eLife*. Your revised article has been evaluated by Tadatsugu Taniguchi (Senior Editor) and a Reviewing Editor.

The manuscript has been improved but there is a small remaining concern that one of the Reviewer would like to see addressed, as outlined below:

"The concern with the killing assay is that the 24 and 48h time points at which the remaining CTV-labeled target cells were measured do not take cell proliferation of T cells or targets into account. Thus, the increased differences at 48h compared to 24h could be skewed by differential cell proliferation. A control would be needed to account for this."

---

## [Author Response]

The work primarily uses the Jurkat T cell line and demonstrates that eIF3 binds to TCRA and TCRB RNAs. The 3' UTR of TCRA and TCRB was responsible for eIF3 binding and this interaction resulted in a rapid burst in TCRA and TCRB translation through a mechanism dependent on CD28. Adding the TCRA or TCRB 3' UTR to a CAR-T construct improved its ability to kill tumor cells. Although the work has potential for publication in eLife, it requires essential additional data to support the central claims of the paper.

We thank the reviewers for their generally positive view of the results we present in the manuscript. We address their concerns as described in more detail below. We also include additional experiments, also as described below.

Each reviewer raised substantive concerns (see below) that need to be resolved experimentally. To quote a few, you should document in Jurkat and in primary T cells whether the rapid burst in TCRA and TCRB translation results in TCR up regulation on the cell surface. This would be important for the functional implications of your findings and considering that (1) CD247 is generally considered as the limiting factor in TCR surface expression, and (2) primary T-cells display a phenomenal sensitivity towards antigen, as the presence of typically only a few antigens on the APC is sufficient to trigger the full T-cell response.

These are interesting points which we now address in more detail in the Discussion. Although we did not look at surface expression of the TCR directly, we did look at TCR clustering after activation with anti-CD3/anti-CD28 antibodies (Figure 5C and Figure 5—figure supplement 1C), which requires TCR surface expression as a prerequisite. We saw clear delays in TCR clustering due to the mutations in the *TCRA* and *TCRB* 3’-UTRs. We also looked at TCR levels on the cell surface by FACS after activation by PMA/Ionomycin in Figure 5—figure supplement 1D-1E, again observing a decrease in cell surface TCR due to the mutations in the *TCRA* and *TCRB* 3’-UTRs. We used PMA/Ionomycin because we could not examine TCR levels on the cell surface by FACS after anti-CD3/anti-CD28 activation, for the technical reason that the anti-CD3 antibody we used for activation in all of our assays (clone UCHT1) interfered with binding of the fluorescent antibody we used for TCR detection in FACS experiments.

Along the same line the readout of T-cell activation shown in Figure 5 and 6 are rather weak. For instance, in Figure 6, gating for CD25 and CD69 is hard to see in the raw flow cytometry plots. Can this be adjusted with the biexponential axis? How were these gates set? In Figure 5, providing primary where possible will support results represented purely as graphs.

We thank the reviewers for pointing out these issues of clarity with respect to Figure 5 (we assume the mention of Figure 6 was actually for Figure 5 based on the context). We have divided the Figure 5 figure supplements into separate figures, so that the flow cytometry plots can be included without loss of resolution. We also include a more detailed description of how the gates were set in the Materials and methods and figure legends. In brief, we used the TCR KO cells (Δ*TCR)* to set the CD69+ and CD25+ gating, as we were able to obtain knockout editing efficiency of 95% or higher in those cells.

Among the mechanistic issues, one particularly puzzling issue concerns the observation that the CD28-dependent burst in translation requires membrane tethering whereas the CD3-dependent increase does not.

We agree this is one of the more interesting observations we made. We think unraveling mechanisms underlying the CD28-dependent and membrane-proximal signaling requirement will be an exciting line of experiments to pursue in a follow-up study. We expanded the Discussion section of this observation to highlight this surprising finding, as described in more detail below.

Please note that during the Reviewer discussion, it has been noted that the Western blots displayed in Figure 6D showed strikingly similar bands for donors 1 and 2 in several of the rows probed for CAR-TCR or HSP90. Even the slopes and shapes of these band seem similar. Accordingly, Reviewers have suggested to include the original blots in the revision.

We thank the reviewers for checking these gels. We had included the original blots in the Source Data File 1 zip file. We applied only two adjustments to these gel images for presentation in Figure 6. First, we adjusted the background by setting a “floor” pixel count, and then used linear adjustments to set the maximal values. Second, we rotated the blots so that the various sections are as close to horizontal as possible.

Reviewer #1:Da Silva et al. demonstrate via biochemical and functional assays and bioinformatics that the 3'UTR of both TCRalpha and β are targeted as part of translationally active polysomes by the eukaryotic initiation factor 3 (elf3) in a fashion that depends on the activation of the costimulatory CD28 signaling pathway. They also show by means of mutagenesis of the 3'UTRs that translation of the resulting TCR mRNAs is increased under conditions where elf3 is allowed to bind. By means of nanoluciferase-based reporter assays they also provide evidence that 3'UTR-targeting by elf3 depends on membrane-proximal CD28 signaling. Furthermore, they were able to link activation-driven burst of TCR-translation levels to elf3-3'UTR binding. They present a convincing case that elf3-3'UTR binding in response to TCR-mediated stimulation promotes TCR-clustering, upregulation of CD25 and CD69 (in CD8^+^ T-cells) as well as IL-2 (and to some extent IFNγ) secretion. Lastly, they show that CD19-CARs equipped with either a TCRa- or a TCRb-3'UTR are earlier and more robustly translated upon CAR-T-cell activation with a B-cell lines. This behavior coincides with a somewhat increased CAR-T-cell-mediated killing activity especially at lower CAR-T-cell:target cell ratios or when CAR-T-cells are confronted with a target cell line expressing lower levels of CD19. Overall the study highlights the importance of the 3'UTR for TCR and CAR translation in a setting of CD28-signaling with implications for (CAR-)T-cell functionality.

We thank the reviewer for their overall positive comments on the manuscript. We do note that we see upregulation of CD25 and CD69 in both CD8^+^ and CD4^+^ T cells (Figure 5—figure supplement 2A-2B), although this may not have been clear in how we wrote the figure legend. We have divided those figure supplements into separate files, based on feedback from the reviewers as described above.

StrengthsApart from the overall subject being highly relevant for a better understanding of T-cell physiology, T-cell antigen recognition and the design and implementation of improved CAR-T-cell-based immunotherapies, the study excels with the quality of experimentation and the quantitative nature, its readability and clear conclusions that are in my view nuanced enough to avoid overinterpretation of the findings. To my best knowledge methodologies and the execution of the experiments are all described with necessary clarity and precision leaving little to no room for ambiguities, so that others in the field will be able to reproduce the data.I find both the reasoning underlying the flow of experimentation and the conclusions drawn from individual observations convincing and compelling. Leads from bioinformatics are verified by detailed biochemistry with clean and clear-cut results, which are then followed up with (i) functional studies concerning mRNA-stability, -elf3 binding and -translation and (ii) studies pertaining to T-cell physiology, all of which provide insights of high relevance.The highly quantitative nature of all readouts applied (which all include meaningful negative controls) leaves little to no doubts with regard to the conclusions drawn.

We are gratified the reviewer was satisfied with the rigor of our experiments, and their interest in our results.

WeaknessesI find only few weaknesses which pertain to the (i) proposed mechanism by which CD28-downstream signaling affects elf3-binding to the 3'UTRs and (ii) the readout of TCR-surface expression and also T-cell activation, the latter lacking definition with regards to TCR-/CAR-proximity. Keeping the scope of the study on in mind, I am of the opinion that these subjects warrant in large part a more detailed discussion and in some cases a few more experiments which could be done with reasonable effort.(i) Worthy of discussion: To me it is not clear how the CD28/PI3K pathway affects elf3-3'UTR binding. To what extent the binding elf3 to other parts of the mRNA effected by this pathway? Is this a general effect of one that is specific to selected 3'UTRs. How could this be addressed in future experiments? It has been reported that the PD1-PD-L1 interaction as it often occurs in the tumor microenvironment or in another setting of T-cell exhaustion affects predominantly CD28 signaling. Is the PD1-mediated inhibition of the T-cell response directly linked to a loss in elf3-3'UTR binding?

These are all great questions that we are interested in answering in future studies. In particular, we agree with the reviewer that it will be interesting to determine if there is a connection between PD1-mediated inhibition and eIF3-3’-UTR binding. One key point we didn’t make clearly is that, since both the TCRA and TCRB subunits are integral membrane proteins, the CD28- and membrane-dependent regulation may primarily affect the immune-related membrane proteins or secreted proteins that we identified in the PAR-CLIP data (i.e. B2M, CD28, CD3D, HLA-E, LAT in Supplementary File 5 and TAP1 and TAP2 in Supplementary File 3). This could be addressed using the tethered vs. non-tethered Nanoluciferase reporters, with 3’-UTRs of other mRNAs we identified in the PAR-CLIP experiment. We have expanded the Discussion to address these questions based on our present knowledge.

(ii) The authors show a boost in TCR and CAR translation that depends on the 3'UTR being able to bind elf3. How is TCR- and CAR- surface expression affected in the course of activation? How is activation-induced TCR- and CAR- downregulation affected? Primary T-cells feature a phenomenal sensitivity towards antigen, as the presence of typically only a few antigens on the APC is sufficient to trigger the full T-cell response. The situation with CARs is different in that they require 100x – 1000x more antigen for stimulation. How is (CAR-)T-cell sensitivity affected by elf3-3'UTR binding? Time-lapsed flow cytometry experiments using IP26/recombinant CD19 (both commercially available) for detection and a simple titration experiment (involving plate-bound OKT3 or CD19 for stimulation) would clarify these sets of questions.

These are all excellent questions for future studies. At present we have only used B tumor cells expressing CD19 for stimulation. We think there are likely to be challenges using plate-bound antibodies to address CAR vs. TCR sensitivity due to the fact that CARs likely do not recapitulate immune synapses. For example, it has been shown that CD19 CARs do not cluster on the T cell surface in immune synapses similarly to TCRs (see ^1^). We have expanded the Discussion to address the difference between CAR and TCR sensitivity, which is likely to be a more general problem beyond the CD19 CARs we used here in proof-of-principle experiments. We will explore other options, including possibly using live cell microscopy in the future to tackle these longstanding problems.

I am fond of this manuscript. because it provides an important aspect of TCR- and CAR-related physiology that appears highly relevant for T-cell-related immune protection and pathologies as well as T-cell based immunotherapies. On the one hand, it becomes clear now, that attention will need to be paid when exchanging TCRs via CRISPR-Cas9-based approaches. On the other hand, CARs featuring a TCRa or TCRb 3'UTR may give rise to better overall performance (to be tested in more detail in vitro and in vivo).As I have outlined in previous sections of my review I regard the quality of the presented work as high. I also agree with the flow of experimentation and the conclusions drawn. What is missing in my view is a more detailed investigation (within the scope of this manuscript.) or discussion of the underlying mechanisms (see suggestions in the previous section).

Again, we thank the reviewer for their overall enthusiasm for the manuscript, and interest in the underlying mechanisms. We have expanded the Discussion to address these, and will explore these questions in future studies.

Suggested points:(i) Discuss and assess in more detail how CD28-downstream signaling (PI3K activity) may affect elf3-3'UTR binding.Are Tec-kinases, which are known to boost TCR-downstream signaling, involved? An experiment involving pharmacological Tec-inhibition would be insightful (applying 3'UTR mutants).

We thank the reviewer for pointing out this connection to Tec-kinases. This together with the PD1 experiment described above will be important to address in future experiments. We have added these ideas to the Discussion.

(ii) TCR-/CAR-surface expression: how is it affected by the nature of the TCRa,b 3'UTR before, during and after activation? A simple flow cytometry experiment would tell. For T-cells OKT3 could be used for stimulation and IP26 for surface detection. For CD19-CAR-T-cells, B-cell lines could be used for stimulation and recombinant CD19 for CAR-detection.

As noted above, we did look at TCR clustering as a function of time, which is dependent on TCR surface expression (Figure 5C). We also looked at TCR surface expression after PMA/Ionomycin activation (Figure 5—figure supplement 1C). We also think the fact that CARs do not form clusters like TCRs in the immune synapse needs to be addressed in the actual CAR design itself. This is a general problem with CARs that is beyond the scope of this manuscript. See ^1^. We thank the reviewer for pointing out that OKT3 and IP26 can be used in combination for stimulation and detection of TCR surface expression by flow cytometry. We used anti-CD3 antibody UCHT1, which interfered with IP26 staining in our hands.

(iii) One last aspect that would boost in my opinion the significance of the findings is antigen sensitivity, which could be addressed with little (to reasonable) effort by subjecting (CAR-) T-cells to antigen (OKT3, CD19) immobilized in a titrated fashion onto plates. Readout of activation could be as simple as measuring cytokine secretion in the tissue culture supernatant.

We agree the distinction between CAR affinity and TCR affinity for antigen is an important parameter to consider. However, as noted above due to the differences in how CARs vs. TCRs cluster in the immune synapse, we think these experiments will require a long-term effort that deserves more attention in a follow-up manuscript.

Reviewer #2:The authors explore the role of eIF3 in T cells using a wide variety of experimental techniques. Experiments are very well-explained in the text and documented in the methods. They begin by using PAR-CLIP in the Jurkat T cell line and find that eIF3 binds to TCRA and TCRB, among other RNAs. They also identify a new pattern of eIF3 binding to the coding sequences of many genes, some of which are validated in primary human cells. Looking at the 3' UTR of TCRA and TCRB, the authors show that eIF3 binding to the 3' UTRs is responsible for activation-induced translational bursting, through a mechanism dependent on CD28. Finally, the authors add the TCRA or TCRB 3' UTR to a CAR-T construct and test its impact on CAR expression and cancer cell killing. Some findings in the study are better-supported than others. The finding of eIF3 controlling translational bursting of TCRA and TCRB by binding the 3' UTR is convincing. However, the discovery of pan-mRNA binding behaviour of eIF3 could use more support from primary cells, and the data from CAR-T cell experiments is subtle and low in replicates. For these reasons it is hard to interpret the data.

We thank the reviewer for their interest in our results. In terms of pan-mRNA binding of eIF3 in primary T cells, we were limited by the technical approach of the RNase H experiment. In primary T cells, the *TCRA* and *TCRB* mRNAs have variable 5’-UTRs and coding regions near the 5’-end of the CDS, precluding the design of DNA oligos for targeting RNase H cleavage, and qPCR of the mRNA in this region. We did see the same pattern of eIF3 binding to *DUSP2* mRNA in the RNase H experiments in primary T cells, as observed in Jurkat cells (Figure 2F-2H).

We have now included additional killing assays with the CAR T cells to increase confidence in the cytotoxicity results. Briefly, we propagated CAR T cells in low or high IL2 concentrations (50 U/mL or 500 u/mL, respectively) prior to performing the killing assays, and observed improved killing in the case of CAR mRNAs containing the *TCRA* or *TCRB* 3’-UTRs in both sets of conditions. We also carried out a killing assay using a different CD19 CAR used in the clinic (Kite Pharma) using T cells from one donor. Finally, we made CAR T cells in which we knocked out TCR expression, and also observed improved killing with the *TCRA* or *TCRB* 3’-UTRs with these cells. Overall, we now report killing assays using 6 donors, with a range of CD19 CAR T designs and conditions. This highlights the robust nature of our results in vitro. We have added these additional killing assays in new figure supplements to Figure 6.

1) The use of a cell line in PAR-CLIP discovery experiments detracts from the generalizability of results, particularly the new finding of coding sequence binding. While the validation of several transcripts in primary human cells is appreciated, it is hard to know how common this phenomenon is. It is also unfortunate that TCRA and TCRB could not be validated in primary cells.

We respectfully disagree with this view of the role of the PAR-CLIP experiment. We view the PAR-CLIP experiment primarily as a “hypothesis generator” rather than as a final result. We absolutely agree that validation of the results, using primary cells where possible, is the best approach to interpreting the PAR-CLIP data. We endeavored to take this approach here, and also agree there is still plenty to be explored based on the results from the PAR-CLIP experiment.

2) When looking at transcript localization in Jurkat cells by FISH, all the mRNA transcripts measured were in distinct puncta. Is this common for all transcripts? It would be important to see a transcript that behaves differently in order to conclude that the puncta are significant.

This is an interesting point. We have now compared *GAPDH* mRNA distribution in Jurkat cells activated for 5 hours to *TCRA* mRNA distribution in confocal z-stacks collected in parallel. We used the distance analysis package DiAna in ImageJ (see ^2^) to identify puncta. Surprisingly (to us), we see *GAPDH* mRNA also forming puncta indistinguishable in size distribution compared to *TCRA* mRNA puncta. The number of *GAPDH* mRNA puncta identified using the DiAna segmentation algorithm also matches approximately the number of *GAPDH* mRNAs we expect to find based on transcription profiling (see Supplementary file 4), and comparisons to the absolute number of mRNAs seen in activated B cells (See ^3^). The number of *TCRA* transcripts is lower than *GAPDH*, based on our transcription profiling results (annotated for the constant region *TRAC* in Supplementary file 4). Thus, the “puncta” we originally assumed to be clusters of mRNAs are more likely than not to be individual mRNAs. We have therefore changed the wording in the main text to emphasize that the *TCRA* and *TCRB* mRNAs do not colocalize with P bodies or stress granules, and eliminated any mention of puncta. Combined with the observation that the mRNAs exhibit the pan-mRNA crosslinking pattern with eIF3, this still supports the model that the mRNAs are being actively translated.

3) CAR T cell data is one representative of 2 donors, and the killing enhancement seen with TCRA or TCRB 3' UTR addition is mild. More biological replicates would be needed to convincingly show this with statistical testing.

As noted above, we have included additional biological replicates to address these concerns. Overall, we now report killing assays using 6 donors, with a range of CD19 CAR T designs and conditions. We think this is a good sign that the effects of the *TCRA* and *TCRB* 3’-UTRs are robust to varying experimental conditions.

Recommendations for the authors1) It is possible to expand primary cells to 55 million, and this would enable the PAR-CLIP to be carried out on primary cells.

This is something we can consider for the future. We note that the use of 4-thio-U can be toxic to cells (See ^4^). We spent considerable time optimizing concentrations of 4-thio-U for Jurkat cells that limited toxicity before carrying out the PAR-CLIP experiment. See the Materials and methods section, “4-thiouridine optimization for PAR-CLIP experiments”. We would need to control for likely different responses in primary T cells. We also note that we may have missed the appropriate time to crosslink the cells had we not used Jurkat cells to begin with, followed by validation in primary cells. This is a bit of serendipity that we acknowledge.

2) The authors conclude that eIF3 binding is specific to selected transcripts based on the fact that eIF3 cross-linking isn't correlated with mRNA abundance. Are these selected transcripts simply those that are newly transcribed? This would also explain why the eIF3-bound transcripts are highly enriched for immune processes.

This is an interesting idea. We did carry out transcription profiling and have deposited those data along with the PAR-CLIP data in the Gene Expression Omnibus database. They are also quantified in Supplementary file 4. We also compared the transcription profiles to PAR-CLIP results in Figure 1—figure supplement 1F. Unfortunately, we don’t have a time series to compare to, which could be done in the future. We have looked at Nanoluciferase reporter mRNA levels in Figure 3C and Figure 3H. We do see mRNA levels go up, but the translation levels don’t correlate directly with mRNA levels. The “peak” of the translation burst reaches 5x-10x (Figure 3B and Figure 3G) when mRNA levels are only up about 2x-3x. For these reporter mRNAs, even when translation drops off, mRNA levels stabilize or continue to increase.

3) The comparison of PAR-CLIP results between HEK293T and Jurkat cells relies on ranking and thresholding. It would be preferable to do a more formal statistical comparison.

This is an interesting point. The PAR-CLIP experiments in Jurkat cells were done in a different way than those in HEK293T cells, complicating the direct comparison. First, the sequencing depth in the Jurkat cells was much higher, improving the signal to noise in the experiment relative to the HEK293T cells (See Supplementary files 2 and 3). This is why we used ranking and thresholding by crosslink reads per transcript in the Jurkat experiment. We also used a larger proportion of anti-EIF3B-bound beads to affinity purify the crosslinked eIF3 RNPs from Jurkat cells, again to improve the signal to noise of the experiment. We also used a different RNase (MNase in our case) compared to the HEK293T experiment, based on extensive optimization. Finally, the Venn diagram is only a proxy for eIF3-mRNA interactions, as the pattern of crosslinking differs (i.e. the presence of pan-mRNAs in Jurkat cells). We therefore think the present comparison is the best compromise, given the multiple differences in the experiments. Since the Venn diagrams may be a distraction from the key points we make in the paper, we have moved the Venn diagrams to the figure supplement and moved the GO term figure to the main figure, as suggested below.

i) In several places in the Methods there seem to be conflicting descriptions of Jurkat cells being used at 8 x 105 versus 8 x 106 / mL.

Thanks for catching this discrepancy. We have updated the Methods to indicate the correct number of cells (8x 10^5/mL).

4) Gating for CD25 and CD69 is hard to see in the raw flow cytometry plots. Can this be adjusted with the biexponential axis? How were these gates set? Many percentages on gates in Figure 5 —figure supplement 2a are missing.

As noted for Reviewer #1, we have split off the flow images into separate figure supplements so that the resolution is not lost. Now that these are separate figures, we have added percentages that we originally left off for clarity. We also converted the axes to biexponential axes and clearly mentioned in the Materials and methods that we have used TCR KO cells (Δ*TCR)* to set the CD69+ and CD25+ gating, as we were able to obtain knockout editing efficiency of 95% or higher in those cells.

5) The live cell gate in Figure 6 —figure supplement 1b is non-conventional, at least to this reviewer. Can this be explained?

Thanks for this question. We assume the confusion is that there is an empty channel in the 2-D plot, meaning that there is no signal used from the vertical axis used to select live cells.

Reviewer #3:De Silva et al. 2021 present exciting results regarding the role of the eIF3 complex in T cell activation. The authors use PAR-CLIP to identify the binding patterns of eIF3 in Jurkat cells and find that regions within the 3´ UTRs of TCRA and TCRB mRNAs are sufficient to increase their translation and drive T cell activation. Moreover, expression of the TCRA and TCRB 3´UTRs in CAR T cells improves their ability to kill tumor cells, demonstrating the therapeutic potential of their findings.

We thank the reviewer for their enthusiasm for our results.

– Figure 1: Venn diagrams should include data from non-activated Jurkat cells. The overlap between different data sets (eg. EIF3A/C/B vs. EIF3D/L) should also be represented using a Venn diagram and included in this figure to convince the reader that the IPs against the different subunits are representative of what the entire complex is binding to. GO terms for overlapping mRNAs should be moved to the main figure and discussed in more detail in the text.

This is a good point we now emphasize more strongly. We had noted in the main text that eIF3 crosslinked to about 75x fewer mRNAs in non-activated cells compared to activated cells. In non-activated Jurkat cells, there were very few mRNAs enriched in the PAR-CLIP experiment, to the point we can’t make Venn diagrams using the same cutoffs as for activated cells. We now note this in the Materials and methods, and the data are included in Supplementary files 2 and 3. This distinction is also captured in Figure 1—figure supplement 1C-1D. In terms of the IP, we used a single well-validated anti-EIF3B antibody (Lee et al., 2015) to pull down the entire eIF3 complex. See Figure 1—figure supplement 1B. Given complications with making these Venn diagrams as noted for Reviewer #2, we think the ones we included are sufficient, without overly interfering with the flow of the paper. As noted above, since the Venn diagrams may be a distraction from the key points we make in the paper, we have moved the Venn diagrams to the figure supplement and moved the GO term figure to the main figure, as suggested. We also discuss some of these mRNAs in the context of CD28 signaling, as noted above.

– All sequencing tracks presented in the manuscript should include a scale in order for the reader to better assess the data.

We apologize, we accidently dropped these from the present version of Figure 1E. We have added these back to Figure 1E, as requested. These are also present in Figure 1—figure supplement 1H.

– When discussing the PAR-CLIP data, the authors state that "eIF3 crosslinking does not correlate with mRNA abundance." Was RNA-seq performed in parallel? These data should be discussed in more detail and the relevant analyses should be included in the main figures.

We did compare mRNA abundance to the PAR-CLIP data in Figure 1—figure supplement 1F. We now note this more clearly in the Results and Materials and methods. The data are included in Supplementary file 4. We prefer to keep the panel in the supplemental figure, which in *eLife* formatting is kept in-line with the text along with the main figure.

– The authors specify that the sites that were modified within the 3´UTRs of TCRA and TCRB were identified through PAR-CLIP. However, the size of the deleted sequence is only mentioned in the context of their CRISPR/Cas9 deletion mutants (Figure 5 supplement 1A). Was the same region targeted in their reporters? Is there evidence that TCRA and TCRB mRNAs that contain these deletions in their 3´UTRs are adequately exported out of the nucleus?

Thanks for pointing out this ambiguity. We did indeed remove (or reverse) the same sites in the 3’-UTRs in the reporter experiments. We did not analyze mRNA export, but it is unlikely that defects in mRNA export can explain our observations. The mRNAs with a deletion or reversal of the 3’-UTR PAR-CLIP sites are still translated (Figure 3B). Thus, if mRNA export were the only problem with these mRNAs, we would still have seen a translational burst from the mutant mRNAs. We don’t observe a burst from these mRNAs despite the fact they are translated, arguing for a direct effect of the 3’-UTR deletions/reversed sequences on translation. Additionally, we don’t see appreciable differences in mRNA levels in Figure 3C with the deletions or reversed sequences compared to WT. In fact, in the cases of *TCRA ΔPAR* and *TCRA* R**PAR* mRNA levels were actually higher than WT.

– The data presented in figure 2 nicely demonstrate that eIF3 is bound to the coding regions of certain mRNAs during their translation. However, it does not provide evidence that the binding of eIF3 occurs "via" the coding sequence as the authors claim in the Discussion section of the manuscript.

We agree this is perhaps an ambiguous wording, and doesn’t capture our findings with respect to the 3’-UTR. We have changed the wording to indicate that eIF3 binding “includes the mRNA coding sequence.”

– The lack of colocalization with stress granules and P bodies alone does not provide convincing evidence that the TCRA and TCRB are in "translational hot spots." Figure 1F should be moved to the supplements as supporting information for the sucrose gradients presented in figure 2.

We have qualified the wording to soften the interpretation: “These results suggest… and are possibly…” We think the figure panel fits better in Figure 1, given this new wording.

– In figure 3E, the authors demonstrate that the 3´UTRs of TCRA and TCRB are required for the interaction between these RNAs and eIF3. What is the proposed mechanism for interactions between eIF3 and the 3´UTR that appear to drive the extensive interaction of eIF3 with the mRNA and the observed translational increase?

This is an excellent question that we hope to address in future work. One set of experiments that we now reference in the Discussion is negative stain EM data on the ER membrane by A. Kent Christensen *et al.*, in which the 5’ and 3’ ends of mRNAs are in close proximity in polysome “circles” or “hairpins”. (See ^5,6^).

– How does eIF3 spread across the CDS? What is the relevance of the interactions between eIF3 and the CDS?

This is a very good question. We noted recent evidence in the Discussion that eIF3 remains on translating ribosomes at least in the initial portion of the CDS. This could be a similar phenomenon to our observations. Future experiments will be needed to sort this out.

– The authors should discuss why the CD28-dependent burst in translation requires membrane tethering but the CD3-dependent increase does not. This section emerged from nowhere and greater explanation would be helpful both for the technical details and for the potential physiological importance of the membrane tethering dependency.

We thank the reviewer for these questions, and we have expanded our rationale for pursuing this line of experiment, in the Results and Discussion sections.

– De Silva et al. demonstrate that expressing the 3´UTRs of TCRA and TCRB in CAR T cells leads to a faster increase in CAR protein expression. However, their data indicate that the increase in translation is transient (Figure 6D and Figure 6 supplement 1A). Therefore, the authors should comment on whether this decreased lag time in protein expression might also accelerate CAR T cell exhaustion.

This is a good question, and we have commented in the Discussion that future experiments will be needed to explore the effect of CAR expression on T cell exhaustion.

[Editors' note: further revisions were suggested prior to acceptance, as described below.]

The manuscript has been improved but there is a small remaining concern that one of the Reviewer would like to see addressed, as outlined below:"The concern with the killing assay is that the 24 and 48h time points at which the remaining CTV-labeled target cells were measured do not take cell proliferation of T cells or targets into account. Thus, the increased differences at 48h compared to 24h could be skewed by differential cell proliferation. A control would be needed to account for this."

We thank the reviewer for raising this question. The reviewer is right that we did not consider CAR T cell propagation in our analysis, which could have confounded are results to some extent. Importantly, we started all experiments with the same number of cells at the designated ratios at the beginning of the assay. We also note that due to the staining of the NALM6 and Jeko 1 cells with CTV, we were able to separate the B tumor cells from the CAR T cells in all conditions over time, which we now diagram in Figure 6—figure supplement 1B. This revised panel shows the separation at 24 hrs. We also see similar separation at 48 hrs and 72 hrs. This indicates that the B tumor cells are not replicating appreciably in these conditions, and even for cells that replicate (seen at 48 hrs and 72 hrs as a drop in CTV signal for a subset of cells), we can cleanly separate B tumor cells from CAR T cells in our analysis. See Author response image 1.

**Author response image 1. sa2fig1:** 

We have updated the cytotoxicity bar graphs according to this updated sorting strategy in two ways. First, we report the percent of NALM6 or Jeko 1 cells killed using the ratio of dead B tumor cells to total B tumor cells, as derived from the CTV staining. This analysis eliminates any confounding cell counting effects due to the presence of CAR T cells. Second, we have now added bar graphs showing the relative number of live CAR T cells in each condition. The values are normalized to the number of WPRE CAR T cells in the 1:1 effector:target conditions at each time point. We used this normalization since we are interested in CAR T cell propagation rather than the control cells. Notably, we almost invariably see that CAR T cells expressing the anti-CD19 CAR using the *TCRA* or *TCRB* 3’-UTR elements propagate better than CAR T cells using the *WPRE* or *3’-LTR* elements. This suggests that using the *TCRA* and *TCRB* 3’-UTRs to express the anti-CD19 CAR improves the health of these CAR T cells relative to the *WPRE* and *3’-LTR* variants. We see this as an exciting observation worth analyzing in future experiments, for example to understand the impact of these 3’-UTRs on CAR T cell exhaustion.

The updated figures and associated legends are Figure 6F-6G, Figure 6—figure supplement 1B-1D, Figure 6—figure supplements 3-4, Figure 6—figure supplement 5C-5D, and Figure 6—figure supplement 6B-6C.